# Antibacterial Potential by Rupture Membrane and Antioxidant Capacity of Purified Phenolic Fractions of *Persea americana* Leaf Extract

**DOI:** 10.3390/antibiotics10050508

**Published:** 2021-04-29

**Authors:** Laura María Solís-Salas, Crystel Aleyvick Sierra-Rivera, Luis Enrique Cobos-Puc, Juan Alberto Ascacio-Valdés, Sonia Yesenia Silva-Belmares

**Affiliations:** Department of Food Science and Technology, Faculty of Chemical Sciences, Autonomous University of Coahuila, Saltillo 25280, Mexico; laura.salas@uadec.edu.mx (L.M.S.-S.); crystelsierrarivera@uadec.edu.mx (C.A.S.-R.); luis.cobos@uadec.edu.mx (L.E.C.-P.); alberto_ascaciovaldes@uadec.edu.mx (J.A.A.-V.)

**Keywords:** *Persea americana*, antibacterial effect, action mechanism, antioxidant, chemical characterization

## Abstract

The present research focused on evaluating the antibacterial effect and the mechanism of action of partially purified fractions of an extract of *Persea americana*. Furthermore, both its antioxidant capacity and composition were evaluated. The extract was fractionated by vacuum liquid chromatography. The antimicrobial effect against *Staphylococcus aureus* (ATCC 6538), *Escherichia coli* (ATCC 11229), *Pseudomonas aeruginosa* (ATCC 15442), and *Salmonella choleraesuis* (ATCC 1070) was analyzed by microdilution and the mechanism of action by the Sytox green method. The antioxidant capacity was determined by DPPH, FRAP, and ABTS techniques and the composition by Rp-HPLC-MS. All fractions showed a concentration-dependent antibacterial effect. Fractions F3, F4, and F5 (1000 µg/mL) showed a better antibacterial effect than the extract against the bacteria mentioned. The F3 fraction showed inhibition of 95.43 ± 3.04% on *S. aureus*, F4 showed 93.30 ± 0.52% on *E. coli*, and F5 showed 88.63 ± 1.15% on *S. choleraesuis* and 86.46 ± 3.20% on *P. aeruginosa*. The most susceptible strain to the treatment with the extract was *S. aureus*. Therefore, in this strain, the bacterial membrane damage induced by the extract and fractions was evidenced by light fluorescence microscopy. Furthermore, the extract had better antioxidant action than each fraction. Finally, sinensitin was detected in F3 and cinnamoyl glucose, caffeoyl tartaric acid, and cyanidin 3-*O*-(6′′-malonyl-3′′-glucosyl-glucoside) were detected in F4; esculin and kaempferide, detected in F5, could be associated with the antibacterial and antioxidant effect.

## 1. Introduction

Antimicrobial resistance represents a global public health problem that has been increasing for years. This phenomenon limits the treatment of moderate and severe microbial infections [1]. Antimicrobial resistance occurs when a bacterium produces enzymes such as phosphorylases, acetylases, adenylases, and beta-lactamases that degrade the antibiotic, reducing its plasma concentration [2], and in consequence, the bacteria survive and develop molecular mechanisms that protect them from a particular class of antibiotic [3,4]. *Clostridium difficile*, *Escherichia coli*, *Clostridium perfringens*, *Staphylococcus aureus*, and *Pseudomonas aeruginosa* are examples of drug-resistant bacteria [1,5,6,7]. Currently, around 700,000 people die worldwide from infections caused by antimicrobial-resistant bacteria. Furthermore, the treatment of drug-resistant infections requires special equipment, prolonged stay, and isolation of patients in hospital facilities, by which the cost of treatment is increased considerably. The above has led to the seeking of a new classes of antibiotics [2,3].

Some plant extracts are rich in biologically active compounds with antioxidant capacity [8]. Interestingly, a relationship between the antioxidant and antimicrobial effect of polyphenolic compounds from plants has been previously described [9]. Proanthocyanidin A, isolated from cranberry, is an example, as its best established medical applications are the prevention and treatment of infections of the gastric mucosa and oral cavity. Nowadays, the starting point of many investigations is focused in identifying the antioxidant properties of pure compounds and their interactions [10]. In this sense, recently, it has been shown that the extract from *P. americana* epicarp has an antioxidant capacity and can be used as a functional food ingredient or nutraceuticals with antioxidant and anti-aging activity [11]. In the same line of evidence, some reports have demonstrated that the extract from *Persea americana* leaves has antibacterial, antifungal, and antioxidant properties [12,13,14].

The extract from the leaves of *P. americana* leaf affects the biofilm of *Candida albicans*, *Escherichia coli*, *Staphylococcus aureus*, *Pseudomonas aeruginosa*, *Streptococcus pneumoniae*, *Klebsiella pneumoniae*, *Streptococcus agalactiae*, and *Bacillus cereus*. At least for *P. aeruginosa* and Bacillus cereus, this action could be related to its antioxidant activity. Similarly, the antioxidant and antimicrobial potential against *S. aureus*, *Salmonella* spp., and *Listeria monocytogenes* (MIC ≥ 750 µg/mL) have been reported for avocado (Hass) peel extract, and their fractions isolated from HPLC-qTOF-MS/MS [15,16,17,18].

*P. americana* (Lauraceae) is of American origin and grows in tropical or subtropical areas [19,20] of Mexico and Central America [21]. The fruit is used as food [21], but the leaves are discarded during harvest and have no industrial application [22]. However, the leaves are used as a seasoning to prepare food to enhance flavor and prevent spoilage. Additionally, the leaves are used in traditional medicine for their bioactive compounds, among which polyphenols stand out [23].

The extract obtained from its leaves contains tannins, anthocyanins, flavonoids, kaempferol, alkaloids, triterpenoids, carbohydrates, and fatty acids [12,15,24,25,26], and these compounds are synthesized by secondary metabolism and usually possess complex structures [27]. On this basis, it is clear that leaves from *P. americana* represent a potential source of molecules with antibacterial effects. Therefore, the aim of the present work was to evaluate the antibacterial effect and the effect on bacterial membrane permeability, and the antioxidant effect of partially purified fractions of an extract of *P. americana*. Bacteria used for this purpose correspond to microorganisms of vital medical importance, associated with gastrointestinal and respiratory diseases, which have developed mechanisms of tolerance to conventional antibiotics [28,29,30].

## 2. Results

### 2.1. Extraction and Fractionation of P. americana Leaf

The yield of the extract of *P. americana* leaves recovered in 2 h was 6.9 ± 0.9%. The fractionation of the extract employing vacuum liquid chromatography (VLC) using a gradient elution of Hex-EtOAc (100:0 → 0:100) separated 11 fractions (F1–F11) with the different yield and retention factors presented in Table 1. Some fractions exhibited the same TLC separation pattern. Therefore, F6, F7, and F8 were grouped into FM1, while F9 and F10 were grouped into FM2. The fraction F1 showed the highest yield of 36.94%, F11, F2, and F7 showed yields of 12.33, 8.89, and 7.71%, respectively, while F3, F4, F5, F6, F8, F9, and F10 showed less than 5%.

Additionally, an extract recovery of 83.36% was estimated by adding the yield of each fraction. Therefore, 16.64% of the extract was retained at the origin of the chromatographic column.

The fractions obtained were used to evaluate the antibacterial effect and membrane permeabilization. Additionally, the antioxidant effect and the composition of each fraction were evaluated by HPLC.

### 2.2. Antimicrobial Effect and Mechanism of Action

#### 2.2.1. Minimal Inhibitory Concentration (MIC)

Ceftriaxone inhibited, at 100%, the growth bacterial of bacteria analyzed at all concentrations assayed. On the other hand, the highest concentration of the extract (1000 µg/mL) showed a lesser inhibition than ceftriaxone (1000 µg/mL) because it presented a range of 43.2 ± 1.4 to 77.8 ± 1.0% of inhibition on the strains studied. On this basis, the higher concentration of each fraction showed an improved antibacterial action compared to the extract (1000 µg/mL): F3 and FM1 fractions against *S. aureus*; F4 and F5 fractions against *S. cholerasuis*; F4, F5, FM1, and FM2 fractions against *E. coli*; and F5 and FM2 fractions against *P. aeruginosa* (Table 2, Table 3, Table 4 and Table 5).

The F3 and FM1 fraction were the most active on *S. aureus*, with an inhibition similar to ceftriaxone at 1000 µg/mL, followed by FM1, with inhibition values of 95.4 ± 3.0 and 90.5 ± 3.0%, respectively. The bacterial growth inhibition data allowed us to estimate an MIC_90_ = 1215.4 ± 15.4 µg/mL for the extract and 941.9 ± 47.0 and 741.7 ± 7.2 µg/mL for F3 and FM1, respectively. F4, F5, and FM2 showed a better effect than the extract, inhibiting the growth of *S. choleraesuis* and *E. coli*, while F5 and FM2 had the same inhibitory effect on *P. aeruginosa* at 1000 µg/mL. MIC_90_ value for the extract was of 1824.8 ± 41.0 and 1146 ± 8.4µg/mL for *S. choleraesuis* and *E. coli*, whereas F4, F5, and FM2 had a MIC_90_ = 899.8 ± 8.0 and 801.8 ± 9.4, 860.8 ± 28.0 and 789.0 ± 2.0 µg/mL, and 1010.6 ± 16.2 and 890.9 ± 16.7 µg/mL, respectively (Table 6).

*S. choleraesuis* was the strain least susceptible to the extract (43.2 ± 1.4), but its fractionation (F4, F5, and FM2), almost in all cases, duplicated its antibacterial effect. The growth of *P. aeruginosa* was inhibited by F5 and FM2 at 86.5 ± 3.2 and 87.9 ± 2.1% at 1000 µg/mL, respectively. An MIC_90_ = 1451.7 ± 23.9 µg/mL was determined for the extract and 842 ± 27.9 and 804.5 ± 11.9 µg/mL were determined for F5 and FM2, respectively.

#### 2.2.2. Sytox Green Test

The best antibacterial action observed with the extract was against *S. aureus*. Therefore, a membrane permeability assay was performed in this bacterial strain. Samples of *S. aureus* treated with the F1 and F2 fractions, or extract at 1000 µg/mL, showed the highest fluorescence (Extract = F2 > F1). In this assay, the fluorescence indicates bacterial membrane damage. However, our results show that the F2 fraction exhibited a lower inhibitory effect on the growth of *S. aureus* (21.2 ± 2.4%). The above suggests that fluorescence observed in the samples treated with F2 is not due to bacterial membrane damage; we speculate that some compounds (not identified in the present work) anchored to the membrane emits fluorescence. This represents a new line of investigation (Figure 1).

The F1 fraction (78.8 ± 2.5%) and the extract (77.8 ± 1.0%) had the same inhibitory effect on bacterial growth in this strain, and the fluorescence observed in these samples suggests that the bacterial membrane is impaired (Figure 1). On the other hand, the fluorescence observed with the fractions with better antibacterial effect than the extract, such as F3, F4, and F5, was scarce or moderate. This effect is probably caused by the low bacterial growth (95.4 ± 3.0, 82.6 ± 1.0, and 84.9 ± 1.7% of inhibition). Interestingly, the fluorescence detected with the F11 fraction (57.6 ± 1.2) was slightly higher than fractions F3, F4, and F5. Admittedly, we do not have a clear-cut explanation for this effect, although it could be due to its antioxidant actions. Finally, the samples treated with FM1 and FM2 did not emit fluorescence, although FM1 and FM2 had inhibition values of 90.5 ± 3.0 and 75.9 ± 2.5%, respectively. We believe that the antibacterial actions of these fractions could be due to another mechanism different from membrane permeability modulation.

### 2.3. Detection of Antioxidant Effect

At this point, it should be highlighted that FRAP value is a measure of reducing power of a molecule, DPPH value is a measure of the radical scavenging capacity of a molecule, and ABTS value determines the antioxidant capacity of a molecule. On this basis, the extract had the highest antioxidant effect in the FRAP, ABTS, and DPPH tests in a ratio of 92:1, 27:1, and 115:1, respectively, compared with the value obtained of the fraction with the best antioxidant effect (F11). This is probably because the mixture of compounds contained in the extract potentiates this effect. All fractions showed an antioxidant effect by FRAP (Table 7). With the exception of FM1, all fractions scavenge DPPH radicals. F1, F5, FM1, FM2, and F11 fractions showed the ability to scavenge ABTS radicals.

Some of the best inhibitor fractions (F3, F4, and FM1) of bacterial growth display an antioxidant effect in at least two methods. Notwithstanding, the F5 and FM2 fractions with moderate antibacterial effect exhibited an antioxidant effect in all assayed methods; F11, the fraction with the high antioxidant capacity among fractions, did not show a notable antimicrobial effect.

### 2.4. Chemical Identification by Rp-HPLC-MS

Chemical analysis by Rp-HPLC revealed twenty phenolic compounds in the separated fractions of the extract. The mass analysis revealed that the fractions contained compounds from families such as hydroxycinnamic acids, lignans, dihydrochalcones, anthocyanins, methoxyisoflavones, hydroxycoumarins, stilbenes, flavonols, thyrosols, and methoxycinnamic acid dimers. Table 8 shows the compounds detected in the separated fractions of the extract of *P. americana* leaf, retention time, and *m*/*z*. Figure 2 shows the chemical structures of the compounds detected by Rp-HPLC.

## 3. Discussion

The extraction procedure (2 h) developed in the present work reduced, by 24 times, the hours spent previously with a better yield (44.74% vs. 15.42%) [30,31]. Additionally, the extraction performance can be improved by employing three sequential extractions every 2 h.

TLC eluted with hexane-ethyl acetate (75:25) allowed us to control the partial purification of the recovered fractions by VLC, because it showed a separation between them with retention factors (Rf) ≥ 0.1 [32]. The VLC showed that the yield of each fraction is due to the chemical composition and the equilibrium of the phases (mobile and stationary) [33]. Although VLC is used to separate extracts from other plants, our study describes the yield of separated fractions for the first time [34,35].

Bacterial inhibition induced by the extract and fractions is clearly related to the concentration and their chemical composition [36]. Exposure to a chemical agent is usually detrimental at high doses because the biological systems produce adaptive effects [37]. In this sense, *P. aeruginosa* and *S. choleraesuis* show the lowest susceptibility to non-fractioned extract because their growth was inhibited at concentrations above 1400 µg/mL. Additionally, the formation of compact biofilm could conferrer them a major resistance that protects them from the antibacterial properties of the extract [38,39].

In the same direction, MIC_90_ estimation allowed us to determine the susceptibility of the bacteria treated with the extract and fractions [40,41]. Some fractions presented an MIC_90_ ≤ 1000 µg/mL. The above reveals that the antibacterial effect of an extract can be improved through fractionation processes. The results of this study are promising because some fractions inhibited the growth of bacteria of medical importance, such as *S. aureus*, *S. choleraesuis*, and *E. coli*, which are responsible for gastrointestinal diseases, and *P. aeruginosa*, which is responsible for respiratory diseases [28,29,30,42]. At this point, it must be emphasized that a higher concentration of these fractions or extract does not necessarily induce a better antibacterial effect; indeed the MIC value for ceftriaxone is 32 µg/mL, and it is normally prescribed by medicals at doses of 0.5 to 1 g per day in moderate to severe respiratory and gastrointestinal infections [43,44].

The sytox method showed that the extract and the active fractions induce pores in the plasma membrane of *S. aureus* because “Sytox Green” fluorogenic dye only penetrates plasma membranes with compromised integrity [9], and it fluoresces when it binds to DNA [45]. This asymmetric cyanine dye has three positive charges and is completely excluded from living eukaryotic and prokaryotic cells [46]. The best antibacterial action against *S. aureus* was obtained with the FM1 fraction; however, bacteria treated with this fraction did not emit fluorescence. A similar antibacterial response was obtained with the treatment of the F3 fraction where a slight fluorescence was observed; comparable responses have been reported for peptides, proteins, and phenolic compounds from other plants [47,48], indicating that the antibacterial mechanism for F3 on *S. aureus* is related to membrane permeabilization [49]. Furthermore, the above observations suggest that the antibacterial effect of FM1 can be mediated by mechanisms not identified in the present work.

Based on our shreds of evidence, we can suggest that bacterial cultures of *S. aureus* treated with SYTOX Green represent an alternative approach to measure bacterial viability and susceptibility to antibiotics [46]. However, other techniques are required to evaluate bacterial death, such as the plate count technique [50], and to establish a correlation with the concentrations of extract or fraction necessary to eliminate the bacteria. Additionally, it would be convenient to carry out the test on *S. choleraesuis*, *E. coli*, and *P. aeruginosa* because some seem to be more resistant to the effect of the extract and fractions. Besides, they could present a mechanism of action other than the rupture of the membrane. Therefore, more work is required to improve the understanding of the inactivation mechanism in the different strains [49].

Antioxidant activity of the plants is usually related to phenolic compounds due to their redox properties because they play a role in the absorption and neutralization of free radicals [51]. Because the reactions that involve antioxidant activity are complex, they should not be analyzed by a single technique [52], therefore, this property of the extract and fractions were investigated by three methods. Our results show that the extract has more antioxidant effect than its fractions and that the F11 fraction had better antioxidant properties than the others. It is known that the extraction is influenced by the solvent, time, and extraction temperature; in addition, the chemical composition and physical characteristics of each sample determine the biological actions of the compounds obtained [8]. The reduced antioxidant capacity of the fractions could be due to the fact that some antioxidant compounds might be retained by covalent bonds into the column. To improve the antioxidant effect of the fractions, the column could be eluted with more polar solvents after elution with Hexane-Ethyl Acetate (0:100). Methanol, water, and dilute weak acid solutions allow isolating more polar compounds [53,54].

All fractions, except FM1, can scavenge DPPH radicals due to their chemical composition [55]. It is likely that the antioxidant properties of each fraction must be a combined action of biologically active compounds, among which polyphenols stand out [8]. Additionally, non-carboxylic phenols and phenolic acids could be responsible for the effect [56,57]. On the other hand, only F1, F5, FM1, FM2, and F11 had an antioxidant effect by the ABTS method because compounds can donate hydrogen atoms or electrons [58,59], and the rest of the fractions lack an antioxidant system against the radical. The F11 fraction had the more antioxidant effect, although all fractions can reduce Fe^3+^ to Fe^2+^ in the presence of 2,4,6-tripyridyl-s-triazine [60]. The reduction is accompanied by the formation of a colored Fe^2+^ complex [60]. In any case, it seems to be that the antibacterial effects of the fractions do not depend on their antioxidant properties.

The Rp-HPLC analysis showed that the fractions contain compounds similar to those of the fruit and epicarp; flavonoids (flavones, flavanones, flavonols, catechins), vanillin, coumarin, procyanidins, phenolic acids, tyrosol-glucosyl-pentoside, pyrocatechol, derivatives of glycosylated acids, and derivatives of phenolic alcohol are examples [23]. Additionally, acetogenins and terpenoid glycosides have been detected [61]. These compounds belong to the hydroxycinnamic acid and hydroxybenzoic acid families. Furthermore, saponins, steroids, terpenes, tannins, alkaloids, flavonoids, terpenoids, and cardiac glycosides have been detected in leaf and seed extracts of *P. americana* obtained with methanol, acetone, and ethyl acetate [62,63]. Although it is clear that the composition depends on the type of extraction.

The antibacterial and antioxidant effects of the fractions separated by VLC are probably related to polyphenolic compounds [64]. In these fractions were detected caffeic acid 4-*O*-glucoside, medioresinol, 3-hydroxyphloretin 2′-*O*-glucoside, cyanidin 3-*O*-glucosyl-rutinoside, sinensitin, cinnamoyl glucose, caffeoyl tartaric acid, cyanidin 3-*O*-(6′′-malonyl-3′′-glucosyl-glucoside), esculin, kaempferide, tangeretin, jaceosidin, 3,7-dimethylquercetin, caffeoyl tartaric acid, resveratrol, kaempferol 3-*O*-(2′′-rhamnosyl-galactoside) 7-*O*-rhamnoside, caffeoyl tartaric acid, 3,4-DHPEA-EA, 5-5′-dehydrodiferulic acid, 6′′-*O*-acetylglycitin, p-coumaroyl tyrosine, and p-coumaric acid 4-*O*-glucoside.

The antibacterial effect of F1 could be associated with medioresinol because it acts against strains sensitive or resistant to antibiotics. Therefore, it could be used as a therapeutic agent for bacterial infections [65]. However, its effect could also be associated with caffeic acid 4-*O*-glucoside and 3-hydroxyfloretin 2′-*O*-glycoside because caffeic acid 4-*O*-glucoside has activity on *B. cereus*, *Bacillus subtilis*, *S. aureus*, *P. aeruginosa*, *E. coli*, and *K. pneumonia* [66].

The antibacterial effect of F2 could be associated with sinensitin and cyanidin 3-*O*-glucosyl-rutinoside because some compounds with phenolic groups affect the stability of the bacterial membrane through the formation of pores due to their interaction with membrane proteins [67,68]. The moderate effect in F3 could be related to sinensitin, while the same effect in F5 could be related to esculin and kaemperide because these compounds have an antimicrobial effect [69,70]. Therefore, sinensitin could hold the main responsibility for the effect on *S. aureus*, given that extracts rich in flavonoids have an antimicrobial effect [71]. In this regard, sinensitin displays an antibacterial action against *P. aeruginosa*, in a concentration-dependent manner.

The antibacterial effect of F4 is likely related to cinnamoyl glucose, caffeoyl tartaric acid, and cyanidin 3-*O*-(6′′-malonyl-3′′-glucoyl-glucoside), but the effect of these compounds individually is unknown. Therefore, they could act synergistically [66]. However, cyanidin 3-*O*-(6′′-malonyl-3′′-glucosyl-glucoside) could be involved in the effect of F4, given that some anthocyanins such as cyanidin-3-glucoside have different properties, such as antioxidant and antimicrobial [64].

The esculin and kaempferide of F5 could be associated with the effect on *P. aeruginosa* because esculin acts against *S. aureus*, *Enterococcus faecalis*, *Salmonella enteritidis*, and *Salmonella typhimurium,* while kaempferide acts against *B. cereus* [69,70].

The resveratrol detected in FM1 could be involved in the antibacterial effect because this compound is a natural antimicrobial agent [72]. The effect of FM2 is probably associated with 3,4-DHPEA-EA because it has an antibacterial effect on *S. aureus* (MIC from 125 to 250 μg/mL) and *Staphylococcus epidermidis* (MIC from 7.81 to 62.5 μg/mL) [73]. As described, FM1 and FM2 show a moderate to a high percentage of bacterial inhibition on *S. aureus*, but their effect is not due to membrane damage because the active compounds could act on microbial DNA [74]. Additionally, the effect of F11 is probably associated with 6″-*O*-acetylglycitin, p-coumaroyl tyrosine, and p-coumaric acid 4-*O*-glucoside, although its antibacterial effects are currently unknown.

In conclusion, the extract obtained from leaves of *Persea americana* exhibits antibacterial effect with the following susceptibility order: *S. aureus* < *E. coli* < *P. aeruginosa* < *S. choleraesuis*. However, the fractions obtained from the extract have higher antibacterial power with similar potency in all the above-mentioned strains. The antibacterial effect of the fractions is probably not related to their antioxidant properties.

## 4. Materials and Methods

### 4.1. Extraction and Fractionation of P. americana Leaf

The *Persea americana* leaves were collected in Tenosique, Tabasco (latitude −91.4225 and altitude 17.4756). The plant was identified taxonomically by the herbarium of the Autonomous University of Nuevo Leon.

The leaves of *P. americana* were previously disinfected with 0.25% sodium hypochlorite and dried at room temperature. Then, they were pulverized to a homogeneous particle size and 100 g of leaf powder were extracted with 1000 mL of absolute ethanol. The extraction was carried out with constant stirring at room temperature for 2 h. The extract was filtered and clarified with Whatman # 4 paper and Whatman G/FA fiberglass. The filtrate was dried on a rotary evaporator at 35 °C, coupled to a recirculator (−10 °C). The extraction was performed in triplicate, and the recovery percentage measured by gravimetry. The extracts were kept frozen at −20 °C until later use.

Fractionation of the extract by vacuum liquid chromatography (VLC) was carried out in a 4.3 cm diameter glass column previously packed with silica 60 (Merck). Therefore, 20.8 g of the extract was placed over the packing of the column. Elution was performed by connecting the column to a 20 psi negative pressure source (Felisa F-1500L vacuum pump), using a flow rate of 0.2 mL/min. The elution gradient used as the mobile phase was 200 mL of Hex-EtOAc (100:0 → 0:100). During the fractionation, eleven fractions (F1, F2, F3, F4, F5, F6, F7, F8, F9, F10, and F11) were recovered. Fractions were collected by volume of 200 mL. The fractions were then pooled with the same TLC separation pattern. The separation pattern of the VLC-eluted fractions was monitored on Merck 60 TLC plates (2.5 × 5 cm) and 2000 µg/mL solutions of the extract and fractions. TLC was eluted with 3 mL of Hex-EtOAc (75:25) because the separation with the eluent presents a retention factor (Rf) ≥ 0.1 between fractions [75]. Fractions were detected at 365 nm with ultraviolet light.

The fractions were dried on a rotary evaporator at 30 °C, and Y (%) was determined by gravimetry. Fractions were stored at 4 °C until use.

### 4.2. Antimicrobial Effect and Mechanism of Action

#### 4.2.1. Bacterial Strains and Culture Conditions

Antibacterial effect of the extract from *P. americana* was tested on *Escherichia coli* (ATCC 11229), *Pseudomonas aeruginosa* (ATCC 15442), *Staphylococcus aureus* (ATCC 6538), and *Salmonella choleraesuis* (ATCC 1070) strains provided by the Faculty of Chemistry of the Autonomous University of Coahuila. The above strains were kept at 4 °C on Mueller Hinton agar (Bioxon^®^). From each strain, a bacterial suspension of 1 × 10^6^ CFU/mL was prepared using the Mc Farland scale of 0.5 using Mueller-Hinton broth. Bacterial suppression of each strain was used in minimal inhibitory concentration (MIC) and sytox green methods.

#### 4.2.2. Minimal Inhibitory Concentration (MIC)

To determine the MIC, it was necessary to measure the percentage of inhibition. The broth microdilution method was used in a 96-well plate, according to NCCLS M11-A6 with slight modifications; culture medium without treatment was used as a negative control. The extract, fractions, and the positive control (ceftriaxone) were tested at 0.5, 1.0, 2.0, 3.9, 7.8, 15.6, 31.3, 62.5, 125.0, 250.0, 500.0, and 1000.0 µg/mL. Therefore, serial dilutions (100 µL) were made from a 2000 µg/mL stock solution, and Mueller-Hinton broth and 10% ethanol tested as negative controls. Then, 100 μL of a bacterial suspension of 1 × 10^6^ CFU/mL was added. The microplates were incubated at 37 °C for 24 h, and the absorbance measured at 625 nm in a microplate reader (BioTek Synergy HTX Multi-Mode Reader).

Five repetitions were performed by each experiment, and the data were analyzed by ANOVA. The percentage of inhibition was calculated from the absorbance data using Formula 1 [76]. The minimal inhibitory concentration, 90, was calculated from the linear regression equation [77].
(1)$$Inhibition\;\left(\%\right)=100-\left[\frac{\left(absorbance\;S\right)\left(100\right)}{absorbance\;NC}\right]$$
where: *S* = sample (extract or fraction); *NC* = negative control.

#### 4.2.3. Sytox Green Test

Damage to the bacterial membrane was detected with Sytox [9]. For this, we evaluated the effect of the extract and fractions at 1000 µg/mL in a 96-well microplate. Briefly, 100 µL of extract and fractions and 100 µL of a microbial suspension at 1 × 10^6^ CFU/mL were added per well. Mueller-Hinton culture broth tested as a viability control and Triton X-100 as a positive control. After 24 h of incubation at 37 °C, 5 µL of Sytox^®^ green dye (1 mM) was added to each treatment (nucleic acid staining, Cat. No. S7020) and incubated in the dark for 30 min. Both the bacteria treated with the extract and fractions of *P. americana* and those treated with the positive and viability controls were observed by optical microscopy in a bright field and fluorescence at 100× using a microscope (AxioLab A1, Carl Zeiss, Suzhou, China).

### 4.3. Detection of Antioxidant Effect

The antioxidant capacity of the extract and the *P. americana* leaf fractions were measured by the 2,2-Diphenyl-1-picrylhydrazyl radical scavenging (DPPH) method, the antioxidant power test for iron reduction (FRAP) assay, and the 2,2-azinobis-(3-ethylbenzothiazolin-6-sulfonic acid) radical uptake (ABTS) method.

#### 4.3.1. DPPH Radical Scavenging Method

The antioxidant activity in the extracts and the fractions was evaluated as the DPPH free radical scavenging in a 96-well plate. For the assay, solutions of the extract and fractions were prepared at 1000 µg/mL and a solution of DPPH 6 × 10^5^ mM, dissolved in ethanol (until obtaining an absorbance of 0.700 at 517 nm). Additionally, a TROLOX calibration curve of 5, 10, 25, 50, 125, and 250 µg/mL was prepared. The test consisted of placing 10 µL of extract or fraction, followed by 290 µL of DPPH per well, then incubating for 30 min in the dark at room temperature. Subsequently, the absorbance at 517 nm was read in a microplate reader (TECAN, InfiniteM200PRO) using ethanol as a blank. The radical scavenging capacity was calculated from the calibration curve and expressed as mg equivalents of TROLOX per g of dry weight equivalent fraction (TE mg/g) [60].

#### 4.3.2. ABTS Radical Scavenging Method

The ABTS assay (2,2′-azino-bis (3-ethylbenzothiazoline-6-sulfonic acid)) was adapted to a 96-well microplate and performed with 50 µM ABTS and a TROLOX calibration curve (45, 65, 85, 105, 125, 145, 165, and 185 mg/L). The extract and fractions were prepared at 1000 µg/mL. Therefore, 50 µL of extract, fraction, or concentration of TROLOX were placed in each well independently and mixed with 250 µL of ABTS. The plate was incubated at 37 °C for 15 min, and the absorbance at 414 nm was measured (GEN 5, Biotek Intruments, Inc., Winooski, VT, USA). The radical scavenging capacity was calculated from the calibration curve and expressed as equivalent mg of TROLOX per g of dry weight equivalent fraction (TE mg/g) [60].

#### 4.3.3. Antioxidant Power Test for Iron Reduction (FRAP)

The FRAP assay was adapted to a 96-well microplate and performed with a calibration curve of FeSO_4_·7 H_2_O (200, 400, 600, 800, and 1000 µM). The extract and fractions were prepared at 1000 µg/mL. The FRAP reagent was prepared by mixing a 10 mM TPTZ solution (dissolved in 40 mM HCl), a 20 mM iron chloride hexahydrate (FeCl_3_·6H_2_O) solution, and a 0.3 M acetate buffer (pH 3.6) in a ratio of 1:1:10 (*v*/*v*/*v*). Therefore, 10 µL of extract, fraction, or concentration of Trolox was placed in each well independently, and 290 µL of FRAP reagent was mixed. The plate was incubated at 37 °C for 15 min, and then absorbance at 593 nm was read in a microplate reader (GEN 5, Biotek Intruments, Inc., Winooski, VT, USA). Additionally, ethanol was used as a blank. The FRAP values expressed in micromoles of iron equivalent per dry weight of the material (µM Fe (II)/g) [78].

#### 4.3.4. Chemical Identification by Rp-HPLC-MS

The compounds contained in the extract of *P. americana* leaves were identified by high-resolution reverse phase liquid chromatography (RP-HPLC-ESI-MS) using a method adapted for phenolic compounds. Briefly, an HPLC chromatograph (Varian) was used. The equipment includes an autosampler (Varian Pro Star 410, Cridersville, OH, USA), a ternary pump (Varian Pro Star 230I, Cridersville, OH, USA), and a PDA detector (Varian Pro Star 330, Cridersville, OH, USA). An ion trap mass spectrometer was used as a detector (Varian 500-MS IT Mass Spectrometer, Cridersville, OH, USA), equipped with an electrospray ionization source. First, a solution of the fraction at 1000 μg/mL in ethanol was prepared and filtered through a 0.45 μm nylon membrane. The compounds of *P. americana* fractions were separated on a Denali C18 column (150 × 2.1 mm, 3 μm, Grace, Cridersville, OH, USA), after injection of 5 μL of the fraction. The oven temperature was maintained at 30 °C. The elution was carried out with a gradient of formic acid (0.2%, *v*/*v*, solvent A) and acetonitrile (solvent B). As an initial gradient, 97% A and 3% B were added for 0–5 min, then a linear gradient of 91% A and 9% B for 5–15 min, followed by a linear gradient of 84% A and 16% of B for 15–45 min, ending with a linear gradient of 50% A and B. The column was then washed and reconditioned. The flow rate was maintained at 0.2 mL/min. The compounds were monitored at 245, 280, 320, and 550 nm during the elution and detected at 280 nm. The fractions eluted injected into the source of the mass spectrometer without splitting automatically. The mass spectra were acquired in the negative ion mode [M − H]^−1^ to obtain the *m*/*z* of each compound. Nitrogen was used as a nebulizing gas and helium as a damping gas. The ion source parameters were a spray voltage of 5.0 kV and capillary voltage and temperature of 90.0 V and 350 °C, respectively. Data were collected and processed using MS Workstation software (V 6.9). The compounds were identified according to the retention times and on the interpretation of the *m*/*z* stored in the chromatographic database. Additionally, the identification was supported by information previously reported in the literature [61]. The compounds detected were related to the antibacterial effect by the information of databases. Figure 3 presents a general scheme of the tests used in the study.

### 4.4. Statistical Analysis

The extract and fraction yields were expressed as the mean and standard deviation, *n* = 3. Antimicrobial and antioxidant effect data were statistically analyzed by ANOVA (one-way and two-way) and Tukey’s tests, *p* < 0.05. The results were expressed as the mean and the standard deviation, *n* = 5 and *n* = 3.

## 5. Conclusions

The study showed that the *P. americana* leaf contains compounds with antibacterial effect on bacteria of medical importance. Therefore, it represents a source for obtaining new antibiotics. Because we know that the effect is associated with antioxidant compounds such as sinensitin in F3 and cinnamoyl glucose, caffeoyl tartaric acid and cyanidin 3-O-(6″-malonyl-3″-glucosylglucoside) in F4; esculin and kaempferide, detected in F5. Additionally, it was evident that the compounds of F3 and F4 damage the cell membrane of *S. aureus*. However, additional trials are required to evaluate other mechanisms of action on *S. aureus* and other strains. Accordingly, once the tests have been carried out to validate its use in humans. 

## Figures and Tables

**Figure 1 antibiotics-10-00508-f001:**
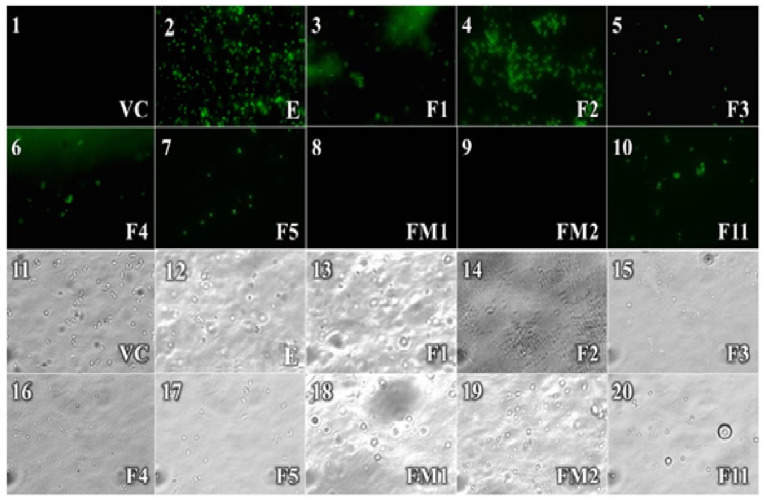
Effect of the extract and fractions of ethanol extract of *P. americana* at 1000 µg/mL on membrane permeabilization on *S. aureus*. Panel 1–10: Observations by fluorescence microscopy of the treatments of the extract and fractions. Panel 11–20: Observations by brightfield microscopy of the treatments of the extract and fractions. VC: viability control; E: Extract; F1-F5, FM1, FM2, F11: Fractions isolated by VLC.

**Figure 2 antibiotics-10-00508-f002:**
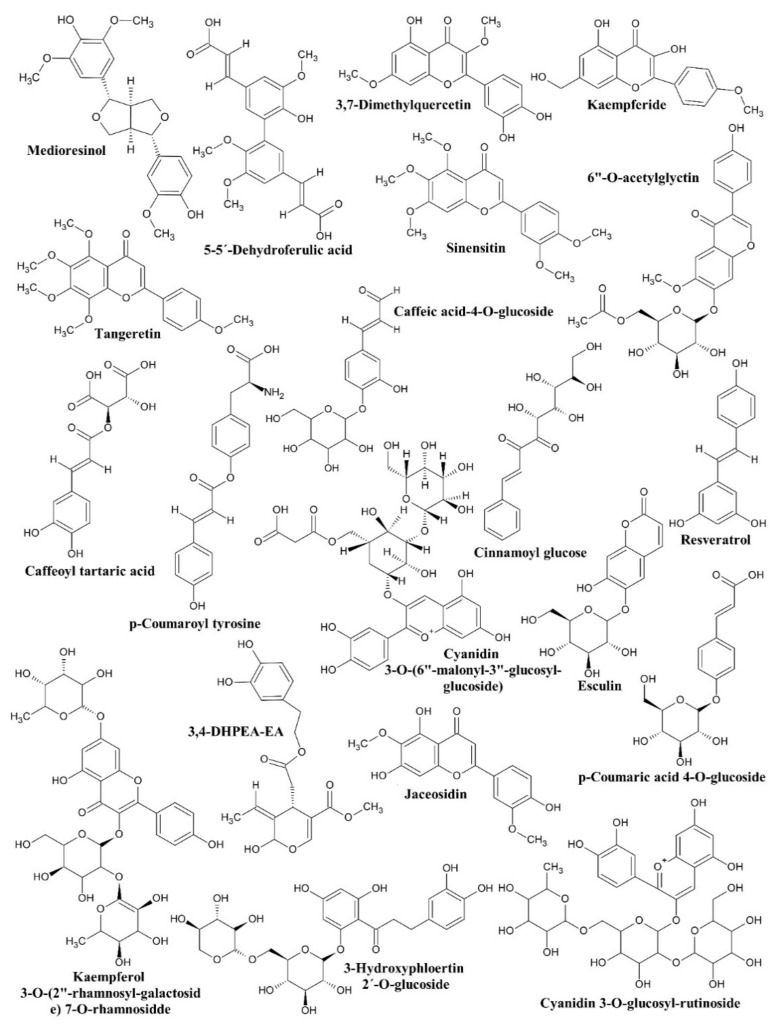
Chemical structures of the compounds detected by Rp-HPLC.

**Figure 3 antibiotics-10-00508-f003:**
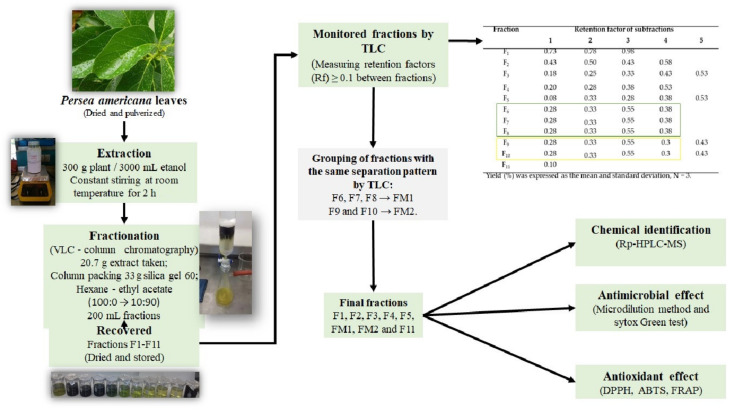
Scheme of the methods used in the study.

**Table 1 antibiotics-10-00508-t001:** Retention factors for VLC fractions.

Fraction	Yield (%)	Retention Factor of Subfractions				
		1	2	3	4	5
F_1_	36.94 ± 0.1	0.73	0.78	0.98		
F_2_	8.89 ± 0.2	0.43	0.50	0.43	0.58	
F_3_	3.48 ± 0.1	0.18	0.25	0.33	0.43	0.53
F_4_	4.12 ± 0.3	0.20	0.28	0.38	0.53	
F_5_	2.77 ± 0.4	0.08	0.33	0.28	0.38	0.53
F_6_	5.58 ± 0.3	0.28	0.33	0.55	0.38	
F_7_	7.71 ± 0.5	0.28	0.33	0.55	0.38	
F_8_	3.39 ± 0.2	0.28	0.33	0.55	0.38	
F_9_	1.64 ± 0.3	0.28	0.33	0.55	0.3	0.43
F_10_	0.51 ± 0.1	0.28	0.33	0.55	0.3	0.43
F_11_	12.33 ± 0.3	0.10				

Yield (%) was expressed as the mean and standard deviation, *n* = 3.

**Table 2 antibiotics-10-00508-t002:** Bacterial inhibition percent of the fractions on *S. aureus.*

µg/mL	Fraction									
	C (+)	Extract	1	2	3	4	5	FM1	FM2	11
0.5	100.0 ± 0.0 ^a^	3.7 ± 1.0 ^a,b^	0.0 ± 0.0 ^a,c^	0.0 ± 0.0 ^a,^^b,c,d^	11.9 ± 1.3 ^b,c,d,e^	4.1 ± 0.7 ^a,d,f^	4.8 ± 1.9 ^a,^^b,c,d,g^	3.3 ± 0.8 ^a,b,c,d,f,g,,h^	1.5 ± 0.2 ^a,d,e,f,g,h,i^	14.5 ± 0.3 ^a,b,c,d,e,f,g,h,i^
1.0	100.0 ± 0.0 ^a^	17.8 ± 1.6 ^a,b^	25.2 ± 3.2 ^a,c^	17.1 ± 1.8 ^a,^^b,c,d^	32.6 ± 0.6 ^b,c,d,e^	31.1 ± 3.0 ^a,d,f^	21.3 ± 2.5 ^a,^^b,c,d,g^	15.0 ± 0.2 ^a,b,c,d,f,g,h^	3.0 ± 1.0 ^a,d,e,f,g,h,i^	31.2 ± 1.3 ^a,b,c,d,e,f,g,h,i^
2.0	100.0 ± 0.0 ^a^	20.4 ± 1.3 ^a,b^	24.9 ± 2.5 ^a,c^	14.9 ± 1.1 ^a,^^b,c,d^	31.7 ± 0.8 ^b,c,d,e^	32.5 ± 1.3 ^a,d,f^	17.5 ± 2.1 ^a,^^b,c,d,g^	19.8 ± 8.0 ^a,^^b,c,d,f,g,h^	14.3 ± 2.0 ^a,d,e,f,g,h,i^	33.1 ± 1.2 ^a,b,c,d,e,f,g,h,i^
3.9	100.0 ± 0.0 ^a^	23.1 ± 0.9 ^a,b^	25.5 ± 3.1 ^a,c^	17.0 ± 1.2 ^a,^^b,c,d^	30.6 ± 1.9 ^b,c,d,e^	32.8 ± 2.11 ^a,d,f^	15.1 ± 1.0 ^a,^^b,c,d,g^	23.1 ± 0.6 ^a,b,c,d,f,g,h^	12.7 ± 1.8 ^a,d,e,f,g,h,i^	33.3 ± 0.7 ^a,b,c,d,e,f,g,h,i^
7.8	100.0 ± 0.0 ^a^	21.6 ± 0.9 ^a,b^	26.3 ± 3.4 ^a,c^	18.1 ± 2.8 ^a,^^b,c,d^	17.4 ± 2.1 ^b,c,d,e^	28.9 ± 1.5 ^a,d,f^	12.6 ± 0.1 ^a,^^b,c,d,g^	15.1 ± 3.8 ^a,b,c,d,f,g,h^	8.1 ± 0.7 ^a,d,e,f,g,h,i^	35.4 ± 4.1 ^a,b,c,d,e,f,g,h,i^
15.6	100.0 ± 0.0 ^a^	24.6 ± 4.3 ^a,b^	23.3 ± 2.4 ^a,c^	18.3 ± 1.8 ^a,^^b,c,d^	29.4 ± 3.4 ^b,c,d,e^	22.3 ± 2.2 ^a,d,f^	15.4 ± 2.3 ^a,^^b,c,d,g^	21.7 ± 5.1 ^a,b,c,d,f,g,h^	16.3 ± 3.1 ^a,d,e,f,g,h,i^	32.8 ± 3.6 ^a,b,c,d,e,f,g,h,i^
31.3	100.0 ± 0.0 ^a^	41.9 ± 1.7 ^a,b^	24.9 ± 4.2 ^a,c^	15.2 ± 1.7 ^a,^^b,c,d^	28.8 ± 2.8 ^b,c,d,e^	24.5 ± 3.1 ^a,d,f^	25.5 ± 1.9 ^a,^^b,c,d,g^	39.3 ± 3.4 ^a,b,c,d,f,g,h^	27.9 ± 3.3 ^a,d,e,f,g,h,i^	37.3 ± 0.5 ^a,b,c,d,e,f,g,h,i^
62.5	100.0 ± 0.0 ^a^	46.1 ± 2.3 ^a,b^	26.4 ± 4.3 ^a,c^	10.0 ± 2.8 ^a,^^b,c,d^	30.7 ± 1.9 ^b,c,d,e^	33.7 ± 1.1 ^a,d,f^	42.9 ± 0.6 ^a,^^b,c,d,g^	51.2 ± 1.5 ^a,b,c,d,f,g,h^	27.0 ± 2.9 ^a,d,e,f,g,h,i^	36.8 ± 2.4 ^a,b,c,d,e,f,g,h,i^
125.0	100.0 ± 0.0 ^a^	46.1 ± 2.3 ^a,b^	67.3 ± 4.3 ^a,c^	46.4 ± 1.1 ^a,^^b,c,d^	50.6 ± 0.8 ^b,c,d,e^	73.9 ± 0.8 ^a,d,f^	41.6 ± 0.8 ^a,^^b,c,d,g^	74.8 ± 1.2 ^a,b,c,d,f,g,h^	41.7 ± 1.3 ^a,d,e,f,g,h,i^	45.0 ± 0.6 ^a,b,c,d,e,f,g,h,i^
250.0	100.0 ± 0.0 ^a^	46.1 ± 2.3 ^a,b^	55.2 ± 3.8 ^a,c^	16.4 ± 3.5 ^a,^^b,c,d^	23.8 ± 1.5 ^b,c,d,e^	71.7 ± 0.3 ^a,d,f^	79.2 ± 0.7 ^a,^^b,c,d,g^	78.5 ± 0.6 ^a,b,c,d,f,g,h^	67.6 ± 2.3 ^a,d,e,f,g,h,i^	36.2 ± 0.6 ^a,b,c,d,e,f,g,h,i^
500.0	100.0 ± 0.0 ^a^	52.7 ± 2.4 ^a,b^	56.4 ± 6.2 ^a,c^	82.1 ± 0.9 ^a,^^b,c,d^	60.9 ± 6.4 ^b,c,d,e^	80.5 ± 1.7 ^a,d,f^	74.4 ± 0.3 ^a,^^b,c,d,g^	87.4 ± 3.2 ^a,b,c,d,f,g,h^	65.9 ± 2.1 ^a,d,e,f,g,h,i^	30.1 ± 0.8 ^a,b,c,d,e,f,g,h,i^
1000.0	100.0 ± 0.0 ^a^	77.8 ± 1.0 ^a,b^	78.8 ± 2.5 ^a,c^	21.1 ± 2.4 ^a,^^b,c,d^	95.4 ± 3.0 ^b,c,d,e^	82.6 ± 1.0 ^a,d,f^	84.9 ± 1.7 ^a,^^b,c,d,g^	90.5 ± 3.0 ^a,b,c,d,f,g,h^	75.9 ± 2.5 ^a,d,e,f,g,h,i^	57.6 ± 1.2 ^a,b,c,d,e,f,g,h,i^

C(+) = ceftriaxone. The inhibition percent is expressed as the mean and standard deviation, *n* = 5. Data were analyzed by ANOVA and Tukey tests, *p* < 0.05. The same letters indicate the significant difference between them.

**Table 3 antibiotics-10-00508-t003:** Bacterial inhibition percent of the fractions on *S. choleraesuis.*

µg/mL	Fraction									
	C (+)	Extract	1	2	3	4	5	FM1	FM2	11
0.5	100.0 ± 0.0 ^a^	0.0 ± 0.0 ^a,b^	1.6 ± 0.7 ^a,b,c^	0.0 ± 0.0 ^a,b,c,d^	0.0 ± 0.0 ^a,b,c,e^	0.0 ± 0.0 ^a,b,e,d,e,f^	0.6 ± 0.0 ^a,b,e,d,e,g^	0.0 ± 0.0 ^a,b,d,e,f,g,h^	2.32 ± 0.9 ^a,b,c,d,e,f,g,h,i^	0.0 ± 0.0 ^a,c,d,e,f,g,h,i^
1.0	100.0 ± 0.0 ^a^	0.0 ± 0.0 ^a,b^	2.4 ± 0.7 ^a,b,c^	0.0 ± 0.0 ^a,b,c,d^	1.4 ± 0.0 ^a,b,c,^	2.18 ± 0.9 ^a,b,e,d,e,f^	1.6 ± 0.1 ^a,b,e,d,e,g^	3.7 ± 0.2 ^a,b,d,e,f,g,h^	13.53 ± 1.9 ^a,b,c,d,e,f,g,h,i^	5.5 ± 0.8 ^a,c,d,e,f,g,h,i^
2.0	100.0 ± 0.0 ^a^	0.0 ± 0.0 ^a,b^	4.7 ± 0.0 ^a,b,c^	0.0 ± 0.0 ^a,b,c,d^	4.4 ± 0.4 ^a,b,c,e^	3.2 ± 0.7 ^a,b,e,d,e,f^	2.1 ± 0.5 ^a,b,e,d,e,g^	7.6 ± 1.5 ^a,b,d,e,f,g,h^	14.14 ± 0.6 ^a,b,c,d,e,f,g,h,i^	10.2 ± 1.7 ^a,c,d,e,f,g,h,i^
3.9	100.0 ± 0.0 ^a^	0.0 ± 0.0 ^a,b^	5.2 ± 0.1 ^a,b,c^	0.9 ± 0.6 ^a,b,c,d^	9.1 ± 0.4 ^a,b,c,e^	4.6 ± 0.1 ^a,b,e,d,e,f^	3.9 ± 0.4 ^a,b,e,d,e,g^	8.9 ± 0.2 ^a,b,d,e,f,g,h^	10.18 ± 0.8 ^a,b,c,d,e,f,g,h,i^	11.7 ± 0.3 ^a,c,d,e,f,g,h,i^
7.8	100.0 ± 0.0 ^a^	0.0 ± 0.0 ^a,b^	5.8 ± 1.0 ^a,b,c^	6.3 ± 0.6 ^a,b,c,d^	10.8 ± 0.3 ^a,b,c,e^	5.7 ± 0.3 ^a,b,e,d.e,f^	6.1 ± 0.1 ^a,b,e,d,e,g^	11.5 ± 0.4 ^a,b,d,e,f,g,h^	15.58 ± 0.7 ^a,b,c,d,e,f,g,h,i^	16.0 ± 0.6 ^a.,c,d,e,f,g,h,i^
15.6	100.0 ± 0.0 ^a^	0.0 ± 0.0	7.4 ± 0.3 ^a,b.c^	7.6 ± 0.2 ^a,b,c,d^	11.4 ± 0.4 ^a,b,c,e^	7.9 ± 0.8 ^a,b,e,d,e,f^	8.8 ± 0.6 ^a,b,e,d,e,g^	12.5 ± 0.7 ^a,b,d,e,f,g,h^	16.37 ± 2.3 ^a,b,c,d,e,f,g,h,i^	20.9 ± 0.6 ^a,c,d,e,f,g,h,i^
31.3	100.0 ± 0.0 ^a^	2.3 ± 0.1 ^a,b^	11.6 ± 1.0 ^a,b,c^	9.7 ± 0.4 ^a,b,c,d^	12.7 ± 0. ^a,b,c,e^	10.3 ± 0.6 ^a,b,e,d,e,f^	8.9 ± 0.8 ^a,b,e,d,e,g^	12.1 ± 0.5 ^a,b,d,e,f,g,h^	21.64 ± 0.4 ^a,b,c,d,e,f,g,h,i^	27.6 ± 1.1 ^a,c,d,e,f,g,h,i^
62.5	100.0 ± 0.0 ^a^	18.5 ± 1.5 ^a,b^	14.4 ± 0.1 ^a,b,c^	12.4 ± 0.3 ^a,b,c,d^	21.6 ± 0.6 ^a,b,c,e^	10.8 ± 0.5 ^a,b,e,d,e,f^	14.7 ± 0.1 ^a,b,e,d,e,g^	20.6 ± 0.7 ^a,b,d,e,f,g,h^	25.09 ± 0.3 ^a,b,c,,d,e,f,g,h,i^	31.6 ± 2.4 ^a,c,d,e,f,g,h,i^
125.0	100.0 ± 0.0 ^a^	14.4 ± 2.4 ^a,b^	35.2 ± 0.1 ^a,b,c^	18.0 ± 0.4 ^a,b,c,d^	25.7 ± 0.7 ^a,b,c,e^	15.4 ± 0.4 ^a,b,e,d,e,f^	26.2 ± 1.2 ^a,b.e,d,e,g^	39.3 ± 1.8 ^a,b,d,e,f,g,h^	26.70 ± 0.4 ^a,b,c,d,e,f,g,h,i^	38.0 ± 1.1 ^a,c,d,e,f,g,h,i^
250.0	100.0 ± 0.0 ^a^	28.9 ± 0.0 ^a,b^	40.7 ± 0.3 ^a,b,c^	29.9 ± 0.6 ^a,b,c,d^	26.1 ± 1.1 ^a,b,c,e^	31.7 ± 0.2 ^a,b,e,d,e,f^	55.4 ± 1.2 ^a,b,e,d,e,g^	49.3 ± 0.5 ^a,b,d,e,f,g,h^	39.20 ± 1.6 ^a,b,c,d,e,f,g,h,i^	46.2 ± 1.0 ^a,c,d,e,f,g,h,i^
500.0	100.0 ± 0.0 ^a^	34.6 ± 0.1 ^a,b^	54.5 ± 0.5 ^a,b,c^	45.4 ± 0.3 ^a,b,c,d^	53.6 ± 1.5 ^a,b,c,e^	68.2 ± 0.5 ^a,b,e,d,e,f^	77.1 ± 0.5 ^a,b,e,d,e,g^	66.0 ± 5.3 ^a,b,d,e,f,g,h^	67.13 ± 1.5 ^a.,b,c,d,e,f,g,h,i^	44.4 ± 0.7 ^a,c,d,e,f,g,h,i^
1000.0	100.0 ± 0.0 ^a^	43.2 ± 1.4 ^a,b^	64.3 ± 1.0 ^a,b,c^	50.5 ± 1.3 ^a,b,c,d^	52.5 ± 1.0 ^a,b,c,e^	91.5 ± 1.0 ^a,b,e,d,e,f^	88.6 ± 1.2 ^a,b,e,d,e,g^	64.4 ± 1.4 ^a,b,d,e,f,g,h^	79.8 ± 0.5 ^a,b,c,d,e,f,g,h,i^	47.0 ± 1.0 ^a,c,d,e,f,g,h,i^

C(+) = ceftriaxone. The inhibition percent is expressed as the mean and standard deviation, *n* = 5. Data were analyzed by ANOVA and Tukey tests, *p* < 0.05. The same letters indicate the significant difference between them.

**Table 4 antibiotics-10-00508-t004:** Bacterial inhibition percent of the fractions on *E. coli.*

µg/mL	Fraction									
	C (+)	Extract	1	2	3	4	5	FM1	FM2	11
0.5	100.0 ± 0.0 ^a^	1.0 ± 0.5 ^a,b^	0.0 ± 0.0 ^a,b,,c^	0.00 ± 0.0 ^a,b,c,d^	3.8 ± 0.3 ^a,b,c,d,e^	2.33 ± 1.0 ^a,b,c,d,e,f^	5.5 ± 1.5 ^a,b,c,d,e,f,g^	4.3 ± 1.1 ^a,b,c,d,e,f,h^	0.6 ± 0.0 ^a,b,c,d,e,f,h,i^	6.2 ± 0.8 ^a,b,d,e,f,g,h,i^
1.0	100.0 ± 0.0 ^a^	2.9 ± 0.4 ^a,b^	3.4 ± 0.5 ^a,,b,c^	10.6 ± 0.8 ^a,b,c,d^	10.2 ± 1.3 ^a,b,c,d,e^	9.3 ± 0.5 ^a,b,c,d,e,f^	12.5 ± 0.6 ^a,b,c,d,e,f,g^	13.2 ± 0.3 ^a,b,c,d,e,f,h^	6.5 ± 0.7 ^a,b,c,d,e,f,h,i^	11.4 ± 0.3 ^a,b,d,e,f,,g,h,i^
2.0	100.0 ± 0.0 ^a^	3.0 ± 0.6 ^a,b^	3.6 ± 0.9 ^a,b,c^	12.1 ± 1.1 ^a,b,c,d^	11.4 ± 1.5 ^a,b,c,d,e^	10.7 ± 1.2 ^a,b,c,d,e,f^	11.4 ± 1.1 ^a,b,c,d,e,f,g^	14.2 ± 0.7 ^a,b,c,d,e,f,h^	6.2 ± 0.8 ^a,b,c,d,e,f,h,i^	12.8 ± 1.0 ^a,b,d,e,f,g,h,i^
3.9	100.0 ± 0.0 ^a^	5.3 ± 1.3 ^a^	3.7 ± 1.3 ^a,b^	11.2 ± 1.1 ^a,b,c,d^	9.2 ± 3.3 ^a,b,c,d,e^	12.3 ± 0.9 ^a,b,c,d,e,f^	13.5 ± 0.6 ^a,b,c,d,e,f,g^	14.3 ± 0.5 ^a,b,c,d,e,f,h^	7.2 ± 0.1 ^a,b,c,d,e,f,h,i^	11.5 ± 1.6 ^a,b,d,e,f,g,h,i^
7.8	100.0 ± 0.0 ^a^	6.3 ± 0.7 ^a,b^	6.3 ± 1.9 ^a,b^	6.4 ± 1.6 ^a,b,c,d^	6.4 ± 1.6 ^a,b,c,d,e^	12.1 ± 1.2 ^a,b,c,d,e,f^	15.0 ± 0.4 ^a,b,c,d,e,f,g^	15.1 ± 0.8 ^a,b,c,d,e,f,h^	7.3 ± 0.2 ^a,b,c,d,e,f,h,i^	12.9 ± 1.0 ^a,b,d,e,f,g,h,i^
15.6	100.0 ± 0.0 ^a^	13.1 ± 1.0 ^a,b^	7.7 ± 0.3 ^a,b^	6.8 ± 0.9 ^a,b,c,d^	7.2 ± 1.0 ^a,b,c,d,e^	14.4 ± 0.6 ^a,b,c,d,e,f^	16.6 ± 1.7 ^a,b,c,d,e,f,g^	15.2 ± 0.5 ^a,b,c,d,e,f,h^	7.6 ± 0.1 ^a,b,c,d,e,f,h,i^	12.7 ± 1.2 ^a,b,d,e,f,g,h,i^
31.3	100.0 ± 0.0 ^a^	15.1 ± 0.2 ^a,b^	9.4 ± 0.8 ^a,b^	7.9 ± 0.7 ^a,b,c,d^	9.1 ± 2.1 ^a,b,c,d,e^	14.9 ± 0.6 ^a,b,c,d,e,f^	17.4 ± 0.5 ^a,b,c,d,e,f,g^	30.4 ± 0.6 ^a,b,c,d,e,f,h^	8.4 ± 0.3 ^a,b,c,d,e,f,h,i^	19.1 ± 0.8 ^a,b,d,e,f,g,h,i^
62.5	100.0 ± 0.0 ^a^	40.6 ± 1.8 ^a,b^	11.1 ± 0.8 ^a,b^	12.7 ± 1.5 ^a,b,c,d^	11.7 ± 1.0 ^a,b,c,d,e^	14.2 ± 1.3 ^a,b,c,d,e,f^	31.3 ± 2.7 ^a,b,c,d,e,f,g^	65.2 ± 1.7 ^a,b,c,d,e,f,h^	15.8 ± 0.5 ^a,b,c,d,e,f,h,i^	23.6 ± 0.8 ^a,b,d,e,f,g,h,i^
125.0	100.0 ± 0.0 ^a^	45.6 ± 2.6 ^a,b^	23.6 ± 0.7 ^a,b^	22.2 ± 0.5 ^a,b,c,d^	21.5 ± 2.1 ^a,b,c,d,e^	37.3 ± 2.3 ^a,b,c,d,e,f^	33.5 ± 3.5 ^a,b,c,d,e,f,g^	80.0 ± 5.6 ^a,b,c,d,e,f,h^	25.0 ± 0.9 ^a,b,c,d,e,f,h,i^	31.9 ± 1.0 ^a,b,d,e,f,g,h,i^
250.0	100.0 ± 0.0 ^a^	46.6 ± 0.2 ^a,b^	17.2 ± 4.1 ^a,b^	20.5 ± 0.5 ^a,b,c,d^	22.0 ± 1.2 ^a,b,c,d,e^	53.9 ± 2.5 ^a,b,c,d,e,f^	65.9 ± 1.5 ^a,b,c,d,e,f,g^	81.1 ± 2.8 ^a,b,c,d,e,f,h^	29.3 ± 0.1 ^a,b,c,d,e,f,h,i^	44.7 ± 0. 6 ^a,b,d,e,f,g,h,i^
500.0	100.0 ± 0.0 ^a^	50.2 ± 0.1 ^a,b^	38.4 ± 3.8 ^a,b^	35.3 ± 1.0 ^a,b,c,d^	33.6 ± 3.0 ^a,b,c,d,e^	81.9 ± 2.8 ^a,b,c,d,e,f^	97.0 ± 2.3 ^a,b,c,d,e,f,g^	85.0 ± 1.0 ^a,b,c,d,e,f,h^	80.5 ± 2.0 ^a,b,c,d,e,f,h,i^	48.7 ± 0.6 ^a,b,d,e,f,g,h,i^
1000.0	100.0 ± 0.0 ^a^	65.9 ± 0.9 ^a,b^	47.0 ± 0.5 ^a,b^	38.0 ± 2.0 ^a,b,c,d^	62.6 ± 1.0 ^a,b,c,d,e^	93.3 ± 0.5 ^a,b,c,d,e,f^	85.1 ± 0.6 ^a,b,c,d,e,f,g^	82.6 ± 0.6 ^a,b,c,d,e,f,h^	86.7 ± 1.5 ^a,b,c,d,e,f,h,i^	45.0 ± 1.5 ^a,b,d,e,f,g,h,i^

C(+) = ceftriaxone. The inhibition percent is expressed as the mean and standard deviation, *n* = 5. Data were analyzed by ANOVA and Tukey tests, *p* < 0.05. The same letters indicate the significant difference between them.

**Table 5 antibiotics-10-00508-t005:** Bacterial inhibition percent of the fractions on *P. aeruginosa.*

µg/mL	Fraction									
	C (+)	Extract	1	2	3	4	5	FM1	FM2	11
0.5	100.0 ± 0.0 ^a^	0.0 ± 0.0 ^a,b^	0.0 ± 0.0 ^a,b,c^	0.0 ± 0.0 ^a,b,d^	0.0 ± 0.0 ^a,d,e^	0.0 ± 0.0 ^a,c,d,e,f^	0.0 ± 0.0 ^a,b,c,d,e,f,g^	0.0 ± 0.0 ^a,b,d,e,f,g,h^	0.0 ± 0.0 ^a,b,c,d,e,f,h,i^	0.0 ± 0.0 ^a,b,c,e,f,g,h,i^
1.0	100.0 ± 0.0 ^a^	3.5 ± 1.4 ^a,b^	0.0 ± 0.0 ^a,b,c^	0.0 ± 0.0 ^a,b,d^	0.0 ± 0.0 ^a,d,e^	0.0 ± 0.0 ^a,c,d,e,f^	0.0 ± 0.0 ^a,b,c,d,e,f,g^	0.0 ± 0.0 ^a,b,d,e,f,g,h^	0.0 ± 0.0 ^a,b,c,d,e,f,h,i^	0.0 ± 0.0 ^a,b,c,e,f,g,h,i^
2.0	100.0 ± 0.0 ^a^	5.3 ± 0.4 ^a,b^	0.0 ± 0.0 ^a,b,c^	0.0 ± 0.0 ^a,b,d^	0.0 ± 0.0 ^a,d,e^	0.0 ± 0.0 ^a,c,d,e,f^	0.0 ± 0.0 ^a,b,c,e,f,g^	0.0 ± 0.0 ^a,b,d,e,f,g,h^	0.0 ± 0.0 ^a,b,c,d,e,f,h,i^	0.0 ± 0.0 ^a,b,c,e,f,g,h,i^
3.9	100.0 ± 0.0 ^a^	6.5 ± 0.9 ^a,b^	0.0 ± 0.0 ^a,b,c^	0.0 ± 0.0 ^a,b,d^	0.0 ± 0.0 ^a,d,e^	0.0 ± 0.0 ^a,c,d,e,f^	0.0 ± 0.0 ^a,b,c,d,e,f,g^	0.0 ± 0.0 ^a,b,d,e,f,g,h^	0.0 ± 0.0 ^a,b,c,d,e,f.h,i^	0.0 ± 0.0 ^a,b,c,e,f,g,h,i^
7.8	100.0 ± 0.0 ^a^	8.6 ± 0.4 ^a,b^	0.0 ± 0.0 ^a,b,c^	0.0 ± 0.0 ^a,b,d^	0.0 ± 0.0 ^a,d,e^	0.0 ± 0.0 ^a,c,d,e,f^	0.0 ± 0.0 ^a,b,c,d,e,f,g^	0.0 ± 0.0 ^a,b,d,e,f,g,h^	0.0 ± 0.0 ^a,b,c,d,e,f,h,i^	0.0 ± 0.0 ^a,b,c,e,f,g,h,i^
15.6	100.0 ± 0.0 ^a^	10.2 ± 0.4 ^a,b^	0.0 ± 0.0 ^a,b,c^	0.0 ± 0.0 ^a,b,d^	0.0 ± 0.0 ^a,d,e^	0.0 ± 0.0 ^a,c,d,e,f^	0.0 ± 0.0 ^a,b,c,d,e,f,g^	0.0 ± 0.0 ^a,b,d,e,f,g,h^	0.0 ± 0.0 ^a,b,c,d,e,f,h,i^	0.0 ± 0.0 ^a,b,c,e,f,g,h,i^
31.3	100.0 ± 0.0 ^a^	10.9 ± 0.7 ^a,b^	0.0 ± 0.0 ^a,b,c^	0.0 ± 0.0 ^a,b,d^	0.0 ± 0.0 ^a,d,e^	0.0 ± 0.0 ^a,c,d,e,f^	0.0 ± 0.0 ^a,b,c,d,e,f,g^	0.0 ± 0.0 ^a,b,d,e,f,g,h^	0.0 ± 0.0 ^a,b,c,d,e,f,h,i^	0.0 ± 0.0 ^a,b,c,e,f,g,h,i^
62.5	100.0 ± 0.0 ^a^	14.8 ± 0.6 ^a,b^	0.0 ± 0.0 ^a,b,c^	0.0 ± 0.0 ^a,b,d^	0.0 ± 0.0 ^a,d,e^	0.0 ± 0.0 ^a,c,d,e,f^	0.0 ± 0.0 ^a,b,c,d,e,f,g^	0.0 ± 0.0 ^a,b,d,e,f,g,h^	0.0 ± 0.0 ^a,b,c,d,e,f,h,i^	0.0 ± 0.0 ^a,b,c,e,f,g,h,i^
125.0	100.0 ± 0.0 ^a^	19.8 ± 0.7 ^a,b^	0.0 ± 0.0 ^a,b,c^	0.0 ± 0.0 ^a,b,d^	6.6 ± 1.7 ^a,d,e^	0.0 ± 0.0 ^a,c,d,e,f^	55.4 ± 1.5 ^a,b,c,d,e,f,g^	33.3 ± 0.5 ^a,b,d,e,f,g,h^	39.0 ± 0.5 ^a,b,c,d,e,f,h,i^	0.0 ± 0.0 ^a,b,c,e,f,g,h,i^
250.0	100.0 ± 0.0 ^a^	29.2 ± 0.8 ^a,b^	30.6 ± 0.7 ^a,b,c^	0.0 ± 0.0 ^a,b,d^	18.7 ± 1.8 ^a,d,e^	0.0 ± 0.0 ^a,c,d,e,f^	47.5 ± 1.5 ^a,b,c,d,e,f,g^	31.8 ± 0.7 ^a,b,d,e,f,g,h^	78.8 ± 0.8 ^a,b,c,d,e,f,h,i^	0.0 ± 0.0 ^a,b,c,e,f,g,h,i^
500.0	100.0 ± 0.0 ^a^	37.6 ± 0.2 ^a,b^	43.8 ± 2.7 ^a,b,c^	0.0 ± 0.0 ^a,b,d^	37.9 ± 3.0 ^a,d,e^	44.9 ± 2.4 ^a,c,d,e,f^	77.2 ± 2.0 ^a,b,c,d,e,f,g^	43.9 ± 1.4 ^a,b,d,e,f,g,h^	84.8 ± 1.0 ^a,b,c,d,e,f,h,i^	0.0 ± 0.0 ^a,b,c,e,f,g,h,i^
1000.0	100.0 ± 0.0 ^a^	61.0 ± 1.4 ^a,b^	49.9 ± 4.1 ^a,b,c^	0.0 ± 0.0 ^a,b,d^	56.6 ± 2.6 ^a,d,e^	66.1 ± 1.0 ^a,c,d,e,f^	86.5 ± 3.2 ^a,b,c,d,e,f,g^	46.5 ± 4.0 ^a,b,d,e,f,g,h^	87.9 ± 2.1 ^a,b,c,d,e,f,h,i^	0.0 ± 0.0 ^a,b,c,e,f,g,h,i^

C(+) = ceftriaxone. The inhibition percent is expressed as the mean and standard deviation, *n* = 5. Data were analyzed by ANOVA and Tukey tests, *p* < 0.05. The same letters indicate the significant difference between them.

**Table 6 antibiotics-10-00508-t006:** MIC_90_ of the extract and fractions of *Persea americana* leaves.

MIC_90_ (µg/mL)				
Extract or Fraction	Bacteria			
	*S. aureus*	*S. choleraesuis*	*E. coli*	*P. aeruginosa*
Extract	1215.4 ± 15.4 a	1824.8 ± 41.0 a	1146 ± 8.4 a	1451.7 ± 23.9 a
F1	1070.0 ± 81.5 a	1212.3 ± 12.4 a	1812.1 ± 69.9 a	1588.2 ± 139.0 a
F2	2534.5 ± 91.5 a	1552.2 ± 28.7 a	2354.4 ± 86.0 a	
F3	941.9 ± 47.0 a	1508.6 ± 26.6 a	1510.1 ± 30.8 a	1480.6 ± 32.2 a
F4	862.6 ± 5.9 a	899.8 ± 8.0 a	801.8 ± 9.4 a	1325.1 ± 9.9 a
F5	836.7 ± 18.7 a	860.8 ± 28.0 a	789.0 ± 2.0 a	841.8 ± 27.9 a
FM1	741.7 ± 7.2 a	1124.0 ± 73.7 a	790.7 ± 7.1 a	1587.2 ± 96.9 a
FM2	985.7 ± 20.9 a	1010.6 ± 16.2 a	890.9 ± 16.7 a	804.5 ± 11.9 a
F11	2110.9 ± 63.9	1819.2 ± 51.8	1836.7 ± 33.9	

The MIC_90_ is expressed as the mean and standard deviation, *n* = 5. The same letters indicate the significant difference between them.

**Table 7 antibiotics-10-00508-t007:** Antioxidant effect of the fractions separated by VLC from the extract of *P. americana* leaf.

Fractions	DPPH (mg ET/g)	ABTS (mg ET/g)	FRAP (µM FE(II)/g)
Extract	508.5 ± 2.8 ^a^	223.5 ± 0.9 ^a^	1477.4 ± 3.9 ^a^
1	0.86 ± 0.13 ^a,b^	4.11 ± 0.14 ^a,b^	4.32 ± 0.48 ^a,b^
2	0.05 ± 0.25 ^a,b,c^	0.00 ± 0.00 ^a,b,c^	4.69 ± 0.40 ^a,b,c^
3	0.35 ± 0.12 ^a.b,d^	0.00 ± 0.00 ^a,b,d^	4.78 ± 0.31 ^a,b,d^
4	0.16 ± 0.14 ^a,c,d,e^	0.00 ± 0.00 ^a,c,d,e^	9.37 ± 0.31 ^a,c,d,e^
5	0.27 ± 0.09 ^a,c,f^	2.20 ± 0.12 ^a,c,f^	5.63 ± 0.30 ^a,c,f^
F_M1_	0.00 ± 0.00 ^a,g^	1.74 ± 0.12 ^a,g^	5.01 ± 0.19 ^a,g^
F_M2_	1.24 ± 0.92 ^a,b,c,d,e,f,g,h^	7.02 ± 0.15 ^a,b,c,d,e,f,g,h^	7.01 ± 0.58 ^a,b,c,d,e,f,g,h^
11	4.44 ± 0.02 ^a,b,c,d,e,f,g,h^	8.02 ± 0.03 ^a,b,c,d,e,f,g,h^	16.22 ± 0.28 ^a,bc,d,e,f,g,h^

The results are expressed as the mean ± SD, *n* = 3. The same letters indicate the significant difference between them.

**Table 8 antibiotics-10-00508-t008:** Compounds identified in the separated fractions of the *Persea americana* leaf extract by Rp-HPLC.

Fraction	tr (min)	*m*/*z*, [M − H]^−^	Compound Identified
F1	5.01	341.1	Caffeic acid 4-*O*-glucoside
	4.54	387.1	Medioresinol
	40.47	451.4	3-Hydroxyphloretin 2′-*O*-glucoside
F2	4.49	757.4	Cyanidin 3-*O*-glucosyl-rutinoside
	53.22	371.3	Sinensitin
F3	39.63	371.3	Sinensitin
F4	52.53	309.0	Cinnamoyl glucose
	55.76	311.1	Caffeoyl tartaric acid
	58.0	694.8	Cyanidin 3-*O*-(6′′-malonyl-3′′-glucosyl-glucoside)
F5	55.87	339.1	Esculin
	57.33	297.1	Kaempferide
F6	40.91	371.1	Tangeretin
	52.88	329.1	Jaceosidin
	53.97	329.1	3,7-Dimethylquercetin
	54.98	311.1	Caffeoyl tartaric acid
	56.79	297.1	Kaempferide
FM1	51.08	227.1	Resveratrol
	55.47	739.3	Kaempferol 3-*O*-(2′′-rhamnosyl-galactoside) 7-*O*-rhamnoside
	56.76	311.1	Caffeoyl tartaric acid
FM2	3.02	377.1	3,4-DHPEA-EA
	35.35	385.	5-5′-Dehydrodiferulic acid
F11	37.34	487.4	6′′-*O*-Acetylglycitin
	52.98	327.2	p-Coumaroyl tyrosine
	53.87	325.1	p-Coumaric acid 4-*O*-glucoside

## Data Availability

Not applicable.

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
