# Peer review of "Antibacterial Potential by Rupture Membrane and Antioxidant Capacity of Purified Phenolic Fractions of Persea americana Leaf Extract"

_antibiotics, 2021, doi:10.3390/antibiotics10050508_

Round 1

Reviewer 1 Report

In the Introduction the authors should better describe the updated research on antioxidant and related references mentioned such as:

Durazzo A. Study Approach of Antioxidant Properties in Foods: Update and Considerations. 02/2017; 6(3):17., DOI:10.3390/foods6030017

A graphical scheme of Extraction and fractionation of P. americana leaf should be inserted.

Tables 2-4 should be checked and data should be better described in the text.

Data on antioxidant properties should be better discussed in the text and compared with previous literature data.

Author Response

Response to Reviewer 1 Comments

Point 1: Does the introduction provide sufficient background and include all relevant references? Can be improved

Response 1: The introduction improved. Additionally, relevant antecedents and references on the antioxidant effect included.

Point 2: Does the introduction provide sufficient background and include all relevant references? Answer: Can be improved

Response 2: The introduction improved. Additionally, relevant antecedents and references on the antioxidant effect included.

Point 3: Are the methods adequately described? Answer: Can be improved

Response 3: The methods were adequately described according to the observations of the two reviewers. It was clarified as fractions were collected by VLCSE clarified monitoring using TLC.  In the microdilution method, it was described how the MIC was estimated by linear regression. The 2,2-Diphenyl-1-picrylhydrazyl in the DPPH test was corrected. Additionally, was modified the use of the term scavering. The article included a graphical abstract.

Point 4: Are the results clearly presented? Answer: Can be improved

Response 4: The results of the antibacterial effect were presented more clearly. Tables 2-5 were modified, in which the extract data and the statistical analysis of the tested samples were included. The MIC table of the most active fractions was included. The results of the sytox test were best described. In the image the nomenclature was modified to match the samples tested. Additionally, the description of the antioxidant effect results was improved.

Point 5: Moderate English changes required

Response 5: The corrections in English were made

Point 6: In the Introduction the authors should better describe the updated research on antioxidant and related references mentioned such as:

Durazzo A. Study Approach of Antioxidant Properties in Foods: Update and Considerations. 02/2017; 6(3):17., DOI:10.3390/foods6030017

Response 6: The wording of the Introduction was improved and other references on antioxidants were added including Durazzo A.

Point 7: A graphical scheme of Extraction and fractionation of P. americana leaf should be inserted

Response 7: A graphic scheme of the study methods was inserted, including the extraction and fractionation of the P. americana leaf.

Point 8: Tables 2-4 should be checked and data should be better described in the text.

Response 8: Tables 2-4 were checked, and the data was better described in the text. Data on antioxidant properties were best discussed in the text and compared with the recommended literature.

Reviewer 2 Report

The subject of this manuscript is interesting, and in scope of the journal. However, it needs major improvement.

  1. If MIC was over the studied range of concentrations, why no higher concentration were checked? Especially since the stock solution concentration was twice as high as the highest concentration used in the assay. It seems that MIC for some factions should be in the range of 1000 – 2000 μg/mL, as at the 1000 μg/mL concentration they gave close to 90% inhibition. And how were MIC values of the whole extract determined? In the Experimental it is written that the concentration range for the whole extract was the same as for the fractions, and reported MICs are outside this range.
  2. The results of bacterial growth inhibition should be discussed in more detail in the Results part. Were the same fractions the most active towards all strains?
  3. Sytox green is generally used for or distinguishing dead from live cells, and the membrane rupture of dead cell is not necessarily the cause of cell death (as the authors assume here), it could be a result of necrosis. So, this assay is not a best choice for the study of membrane permeability increase mechanism, as it gives only an indirect indication of possible mechanism. If used, It is important to investigate whether concentrations of the extract (or fraction) required to lyse bacterial membranes correlate with concentrations required to kill bacteria. It should be done for all strains used in the study, as some bacteria species seem to be more resistant to membrane damage and the mechanism of action can be different than membrane rupture. And here this part of the study was done only for Staphylococcus aureus, which is the only one among strains used here with no outer membrane, and it could be very important for the mechanism of action of studied extract and its fractions. So, the conclusions about the possible mechanism of action towards Escherichia coli, Pseudomonas aeruginosa and Salmonella choleraesuis can not be inferred from experiments using only the Staphylococcus aureus strain.
  4. There is no analysis of potential correlation between antimicrobial activity against various strains and the antioxidant activity, as well as between the results of DPPH, ABTS and FRAP assays. Are there any common trends?
  5. There is no data on the concentration of compounds present in various fractions, and some compounds are present in more than one fraction. So, it would be informative to do at least semi-quantitative analysis (i.e. assess the approximate concentration and compare the concentration of the same compounds in various fractions) to better understand which compounds are the active ones in a given fraction, by analyzing the relation between a compound concentration and fraction activity (both antimicrobial and antioxidant).
  6. Subchapter 2.1.2, line 91: it’s not clear which fractiona were similar: F6 to F7 and F8? F6 to F7, and they were different from F8 which was similar to F9 and F10?
  7. The first two sentences in Results and Discussion are exactly the same. Please decide where they should stay and which version should be removed.
  8. Subchapter 2.3.1, lines 122-123: How fractions could be exposed to something? To what? The bacterial concentration was changing? If that was not the intended meaning, please rewrite this sentence.
  9. In Table 6 the title of the first column is not in English.
  10. As the concentration of ethanol in the extraction solvent is given as 10% w/v, it can hardly be called „ethanolic” extract. It would be better to remove the „ethanolic” adjective.
  11. In the caption of Table 7 it is written that the compounds were identified by VLC – are sure it shouldn’t be HPLC-MS?
  12. Sinensetin is written in the table as sinensetin, but in the text as sinensetine and synesetin. Please unify this.
  13. On what basis it is stated that „extraction performance could be optimized using three se- 195
    quential extractions every 2 hours”? (Discussion, lines 195-196) Were any additional tests performed?
  14. The Authors wrote that „The fluorescence revealed by S. aureus against the active fractions is related to the percentage of bacterial inhibition detected in the microdilution test” (Discussion, lines 219-220). Is this a linear correlation?
  15. The part of discussion concerning antioxidant activity needs to be rewritten. There is no trapping of DPPH radical in the assay, it’s scavenged. The same goes for ABTS radical assay. And this paragraph is really hard to understand.
  16. Discussion, line 247: flavones, flavanones, flavonols, catechins are all subgroups of flavonoids, not different groups of compounds.
  17. Generally, the statements that a given compound is responsible for the activity of a given fraction is too strong without the verification with experiments using pure compounds. So, the observed effect „could be”, „probably are”, but not „are”, as it suggests the certainty not possible with reported experiments.
  18. Sinensetin is not an anthocyanin (lines 276-278), it is a flavone.
  19. The title of subchapter 4.2 is not in English.
  20. Please add how the fractions were collected – by time? By volume? By the change of TLC pattern? It’s not clear, as there is a concentration given for TLC analysis which suggests that this analysis was done after the finalization of fractionation.
  21. What was the reason for chosing which bacterial strains were used in this work?
  22. In the formula 1 there is CN, and it should be NC (negative control).
  23. Subchapter 4.4: it’s 2,2-Diphenyl-1-picrylhydrazyl, and it’s scavenged, not captured.
  24. Was the concentration of DPPH really 105 M?! It’s an enormous concentration, usually for this assay milimolar or micromolar concentration are used.
  25. The extensive English revision is needen, as in many places it makes the reading and understanding of the manuscript very difficult if not impossible.

Author Response

Response to Reviewer 2 Comments

Point 1: Extensive editing of English language and style required

Response 1: The corrections in English were made

Point 2: Does the introduction provide sufficient background and include all relevant references? Answer: Can be improved

Response 2:  The wording of the Introduction was improved and other references on antioxidants were added.

Point 3: Is the research design appropriate? Answer: Can be improved

Response 3: The authors consider that the research design is adequate. However, future research could include some trials to better understand the mechanisms of action of the compounds. A finer purification and quantification could also be carried out to identify with more precision the compounds responsible for the antibacterial effect.

Point 4: Are the methods adequately described? Answer: Must be improved

Response 4: The methods were adequately described according to the observations of the two reviewers. Therefore, it was clarified how the fractions collected by VLC and how they was monitored by TLC. In the microdilution method, it was described how the MIC was estimated by linear regression. The 2,2-Diphenyl-1-picrylhydrazyl in the DPPH test was corrected. Additionally, was modified the use of the term scavering. The article included a graphical abstract.

Point 5:  Are the results clearly presented? Answer: Must be improved

Response 5: The results of the antibacterial effect were presented more clearly. Tables 2-5 were modified, in which the extract data and the statistical analysis of the tested samples were included. The MIC table of the most active fractions was included. The results of the sytox test were best described. In the image the nomenclature was modified to match the samples tested. Additionally, the description of the antioxidant effect results was improved.

Point 6:  Are the conclusions supported by the results? Answer: Must be improved

Response 6: The conclusions are included at the end of the discussion and are supported by the results.

Point 7: If MIC was over the studied range of concentrations, why no higher concentration were checked? Especially since the stock solution concentration was twice as high as the highest concentration used in the assay. It seems that MIC for some factions should be in the range of 1000 – 2000 μg/mL, as at the 1000 μg/mL concentration they gave close to 90% inhibition. And how were MIC values of the whole extract determined? In the Experimental it is written that the concentration range for the whole extract was the same as for the fractions, and reported MICs are outside this range.

Response 7: In the antibacterial effect test, the same concentration range was used for the extract and fractions because expected an increase in the bacterial inhibition of fractions after their concentration by VLC. Accordingly, it was confirmed that the most active fractions have a MIC within the range of concentrations tested from 0.5 to 1000 mg / mL. However, the less active ones are above the tested range. Consequently, they could be evaluated to confirm it. MICs were estimated using linear regression, through the equation of the line (y= mx b, r=1), based on the inhibition percentages.

Point 8: The results of bacterial growth inhibition should be discussed in more detail in the Results part. Were the same fractions the most active towards all strains?

Response 8: The results were better explained. The most active fractions were not the same for all strains. Therefore, F3 showed a 95.4% inhibition against S. aureus, F4 95.5% and 93.3% against S. choleraesius and E. coli. FM2 showed an 87.9% inhibition against P. aeruginosa.

Additionally, the most active fractions showed MIC90 lower than 1000 µg / mL, which was the highest concentration evaluated, as shown in table 6 of the document.

Point 9: Sytox green is generally used for or distinguishing dead from live cells, and the membrane rupture of dead cell is not necessarily the cause of cell death (as the authors assume here), it could be a result of necrosis. So, this assay is not a best choice for the study of membrane permeability increase mechanism, as it gives only an indirect indication of possible mechanism. If used, It is important to investigate whether concentrations of the extract (or fraction) required to lyse bacterial membranes correlate with concentrations required to kill bacteria. It should be done for all strains used in the study, as some bacteria species seem to be more resistant to membrane damage and the mechanism of action can be different than membrane rupture. And here this part of the study was done only for Staphylococcus aureus, which is the only one among strains used here with no outer membrane, and it could be very important for the mechanism of action of studied extract and its fractions. So, the conclusions about the possible mechanism of action towards Escherichia coli, Pseudomonas aeruginosa and Salmonella choleraesuis can not be inferred from experiments using only the Staphylococcus aureus strain.

Response 9: The SYTOX assay was performed to observe compromised plasma membranes by the appearance of fluorescence. Therefore, it is not intended to claim that this is the mechanism of action of bacterial death.

If not, show that the extract or fractions damage or compromise the membrane since some studies have suggested that membrane involvement is a main factor in cell inactivation.

According to :

 Roth B.L., Poot M, Yue S.T., Millard P. J. Bacterial Viability and Antibiotic Susceptibility Testing with SYTOX Green Nucleic Acid Stain. Appl Env. Microbiol 1997, 63 (6), 2421–2431.

McKenzie K., Maclean M., Grant M.H, Ramakrishnan P., MacGregor S.J. , Anderson J. G. The Effects of 405 Nm Light on Bacterial Membrane Integrity Determined by Salt and Bile Tolerance Assays, Leakage of UV-Absorbing Material and SYTOX Green Labelling. Microbiology 162 (9), 1680–1688.

Point 10: There is no analysis of potential correlation between antimicrobial activity against various strains and the antioxidant activity, as well as between the results of DPPH, ABTS and FRAP assays. Are there any common trends?

Response 10: There is no correlation between antimicrobial activity and antioxidant activity since they are techniques with different foundations. Therefore, there is a common trend between the data since the results are very close values.

Point 11: There is no data on the concentration of compounds present in various fractions, and some compounds are present in more than one fraction. So, it would be informative to do at least semi-quantitative analysis (i.e. assess the approximate concentration and compare the concentration of the same compounds in various fractions) to better understand which compounds are the active ones in a given fraction, by analyzing the relation between a compound concentration and fraction activity (both antimicrobial and antioxidant).

Response 11: The study focused on detecting the main compounds of each fraction by Rp-HPLC associated with the antibacterial effect. However, it would be helpful to purify them by HPLC and p steal the effectiveness of each one individually, but it was not possible due to the low yield of each fraction.

Point 12: Subchapter 2.1.2, line 91: it’s not clear which fractiona were similar: F6 to F7 and F8? F6 to F7, and they were different from F8 which was similar to F9 and F10?

Response 12: The information about the grouping of fractions according to the pattern shown by TLC was clarified.

Point 13: The first two sentences in Results and Discussion are exactly the same. Please decide where they should stay and which version should be removed.

Response 13: Both sentences were modified

Point 14: Subchapter 2.3.1, lines 122-123: How fractions could be exposed to something? To what? The bacterial concentration was changing? If that was not the intended meaning, please rewrite this sentence.

Response 14: The sentence was rewritten to improve understanding.

Point 15: In Table 6 the title of the first column is not in English.

Response 15: The title of the first column of Table 6 was corrected

Point 16: As the concentration of ethanol in the extraction solvent is given as 10% w/v, it can hardly be called „ethanolic” extract. It would be better to remove the „ethanolic” adjective.

Response 16: The adjective ethanolic was eliminated except for the abstract and introduction to maintaining the understanding of the text.

Point 17: In the caption of Table 7 it is written that the compounds were identified by VLC – are sure it shouldn’t be HPLC-MS?

Response 17: The writing was corrected because the compounds were identified by HPLC.

Point 18: Sinensetin is written in the table as sinensetin, but in the text as sinensetine and synesetin. Please unify this.

Response 18: The word Sinensetin was unified

Point 19: On what basis it is stated that „extraction performance could be optimized using three sequential extractions every 2 hours”? (Discussion, lines 195-196) Were any additional tests performed?

Response 19: We proposed to extract through three sequential extractions because other authors have extracted approximately twice as much in 48 h. So our extraction method was better. The wording of the paragraph was improved

Point 20: The Authors wrote that „The fluorescence revealed by S. aureus against the active fractions is related to the percentage of bacterial inhibition detected in the microdilution test” (Discussion, lines 219-220). Is this a linear correlation?

Response 20: It is about the relationship between bacterial inhibition and the fluorescence emission of the active fractions. However, not all presented it.

Point 21: The part of discussion concerning antioxidant activity needs to be rewritten. There is no trapping of DPPH radical in the assay, it’s scavenged. The same goes for ABTS radical assay. And this paragraph is really hard to understand.

Response 21: The discussion on antioxidant activity was rewritten. Especially in the DPPH and ABTS methods

Point 22: Discussion, line 247: flavones, flavanones, flavonols, catechins are all subgroups of flavonoids, not different groups of compounds.

Response 22: The writing of this line has been corrected. The flavones, flavanones, flavonols, catechins were placed as subgroups

Point 23: Generally, the statements that a given compound is responsible for the activity of a given fraction is too strong without the verification with experiments using pure compounds. So, the observed effect „could be”, „probably are”, but not „are”, as it suggests the certainty not possible with reported experiments.

Response 23: The tenses of the verbs were modified according to this observation.

Point 24: Sinensetin is not an anthocyanin (lines 276-278), it is a flavone.

Response 24: According to the observation, anthocyanin was modified by flavonoid.

Point 25: The title of subchapter 4.2 is not in English.

Response 25: The title of the subchapter was changed to the English language.

Point 26:  Please add how the fractions were collected – by time? By volume? By the change of TLC pattern? It’s not clear, as there is a concentration given for TLC analysis which suggests that this analysis was done after the finalization of fractionation.

Response 26: The requested information was added to the manuscript. Fractions were collected by volume. After elution the fractions were monitored by TLC, and those that had the same patterns were pooled.

Point 27: What was the reason for chosing which bacterial strains were used in this work?

Response 27: The selection of the bacteria evaluated was due to their medical importance since they are associated with gastrointestinal and respiratory diseases. Besides, these bacteria have developed mechanisms of tolerance to conventional antibiotics.

Point 28: In the formula 1 there is CN, and it should be NC (negative control).

Response 28: The nomenclature was modified

Point 29 Subchapter 4.4: it’s 2,2-Diphenyl-1-picrylhydrazyl, and it’s scavenged, not captured.

Response 29: The word captured was replaced by scavering

Point 28: Was the concentration of DPPH really 105 M?! It’s an enormous concentration, usually for this assay milimolar or micromolar concentration are used.

Response 28: The concentration was corrected to millimolar

Point 29: The extensive English revision is needen, as in many places it makes the reading and understanding of the manuscript very difficult if not impossible.

Response 29: The correction of the English language was carried out.

Round 2

Reviewer 2 Report

The revised version of this manuscript is improved, but it needs some improvement before being valid for publication.

Some earlier remarks that were not taken into account in revised version:

  1. It was not explained in the text why if MIC of the whole extract and of some fractions was over the studied range of concentrations, no higher concentrations were checked. Especially since it was possible to obtain a higher concentration: in the text it is stated that the stock solution concentration was 2000 μg/mL.
  2. As the concentration of ethanol in the extraction solvent is given as 10% w/v, it can hardly be called „ethanolic” extract. It would be better to remove the „ethanolic” adjective.

Although the authors in their answer state that this adjective was removed from the text, it’s still present in some instances. If not removing, than maybe change it to „aqueous-ethanolic”? If 90% of the solvent is wate, than the extract is more aqueous than ethanolic.

  1. Although the part of discussion concerning antioxidant activity was rewritten according to my remarks („there is no trapping of DPPH radical in the assay, it’s scavenged. The same goes for ABTS radical assay”), these expressions should be corrected in subchapter 2.4 and the Experimental part as well. It was not written explicitly in my review of the first version of the manuscript, but it is rather obvious that if it is to be corrected in Discussion, and it’s repeated in Experimental, than it should be corrected in all instances.
  2. Please add to the text that the fractions were collected by volume (was it 1 mL? Or less?). (subchapter 4.1)
  3. Please add to the Introduction the reason for chosing which bacterial strains were used in this work – as it was explained in the answer to the reviewer’s comments.
  4. In the experimental part these changes were still not introduced:

- In the formula 1 there is still CN, and it should be NC (negative control).

- Subchapter 4.4: it’s 2,2-Diphenyl-1-picrylhydrazyl, not 2,2-diphenyl-1-picrylhydracil and it’s scavenging, not capture (the latter one was corrected in subchapter 4.4.1, but not in other instances).

And some more comments:

1A. In the text the name „sinensitin” is used, while in Fig. 2 it’s „sinensetin”.

2A. As extraction and fractionation are now described together in subchapter 4.1, please change the numbering of following subchapters accordingly.

3A. The caption of Table 3 moved to under the table (editing issue).

4A. In Subchapter 2.3.2 there are mentions of Figure 2. However, the results described here are shown in Figure 1.

5A. Could you give at least some hypothesis why the antioxidant activity of the extract is three orders of magnitude higher than the activity of all fractions? And it is much higher than the activity of e.g. clove essential oil which is known for its strong antioxidant properties (FRAP of 622 micromoles of Fe per gram, compared with 3000 milimoles per gram reported here for the extract). It’s very unusual, and seems more like some calculation error.

6A. In Discussion (paragraph  concerning Sytox method) there are contradicting statements: 1) However, some fractions have an effect on S. aureus in ways other than membrane permeabilization. For instance, the sample treated with FM1 did not emit fluorescence but presented MIC90 lower than F3., 2) Therefore, it was confirmed that the antibacterial effect could be due to the increase in permeabilization. So, this paragraph needs rewritting. And it should be stressed out that this results are valid only for S. aureus, as no experiments for other strains has been performed.

7A. Line 365: why „Surprisingly”? it is quite a common occurence.

8A. The caption of Figure 3 contains the expression „Graphical abstract”. It’s not a graphical abstract, it’s a scheme.

9A. In FRAP assay there is no radical present, so it can’t be said that The radical scavenging capacity was calculated (lines 481-482).

10A. In the pdf version of revised manuscripts some „μ” in micromoles notation are not visible.

11A. Please check carefully if the units are reported everywhere when needed (e.g. when the MIC90s are discussed).

12 A. In attached pdf file some instances where grammatical corrections are needed are marked by underlining.

Author Response

                                                                                             April 22, 2021

Prof. Dr. Maria Stefania Sinicropi

Guest Editor

Antibiotics

Dear Dr Sinicropi:

The reason for this message is to send the requested modifications to the manuscript: "Antibacterial potential by rupture membrane and antioxidant capacity of purified phenolic fractions of Persea americana leaf extract".  The file contains the corrections requested by reviewer 2 of the journal antibiotics.  The corrected article indicates the changes made to the manuscript.

Response to Reviewer 2 Comments

Point 1: Is the research design appropriate? Answer: Can be improved

Response 1: Some changes were made in the structure of the manuscript for each point in the results to coincide with those proposed in the methodology. Therefore, the extraction process was combined with the fractionation. Additionally, the recovery by VLC and the monitoring by TLC were clearly described.

Regarding the design of the manuscript were made some adjustments in the spaces between paragraphs. Additionally, adjustments were made to the indentation of the paragraphs. Figure 2 was reduced in size to improve the manuscript design. In addition, the MIC table was modified to improve the presentation of the results and its statistical analysis was included. The presentation of Table 7 was improved. Figure 3 moved to the end of the method.

Point 2: Are the methods adequately described? Answer: Can be improved

Response 2:  The methods were better described, in terms of the extraction process, fractionation and antioxidant effect.

Point 3: It was not explained in the text why if MIC of the whole extract and of some fractions was over the studied range of concentrations, no higher concentrations were checked. Especially since it was possible to obtain a higher concentration: in the text it is stated that the stock solution concentration was 2000 μg/mL.

Response 3:  The reason that concentrations above 1000 mg / mL of the fractions were not analyzed was mainly because the active fractions were expected to be within the range. The reason that concentrations above 1000 mg / mL of the fractions were not analyzed was mainly because the active fractions were expected to be within the range. The same concentration range was used for the extract and the fractions because an increase in bacterial inhibition caused by the active fractions was expected after their concentrating by VLC. Therefore, concentrations ≥ 1000 mg / mL were not tested. Consequently, the most active fractions were confirmed to have MICs within the range of concentrations tested from 0.5 to 1000 mg / mL.

Additionally, we did not consider testing higher concentrations of the extract and fractions since conventional antibiotics have MIC ≤ 1000 mg / mL. Cephthiaxone is one example, as it is normally prescribed by doctors at minimum doses of 0.5 to 1 g per day for moderate to severe respiratory and gastrointestinal infections. This antibiotic has MIC = 32 mg / mL. Therefore, it would not make sense to test higher concentrations since it was intended to find fractions with MIC ≤ 1000 mg / mL.

Yoon Soo Park, Jennifer M. Adams-Haduch, Kathleen A. Shutt, Daniel M. Yarabinec, Laura E. Johnson, Ameet Hingwe, James S. Lewis, James H. Jorgensen, and Yohei Doi. Clinical and Microbiologic Characteristics of Cephalosporin-Resistant Escherichia coli at Three Centers in the United States. Antimicrob Agents Chemother. 2012 Apr; 56(4): 1870–1876. doi: 10.1128/AAC.05650-11

http://www.facmed.unam.mx/bmnd/gi_2k8/prods/PRODS/Ceftriaxona.htm

Point 4:  As the concentration of ethanol in the extraction solvent is given as 10% w/v, it can hardly be called „ethanolic” extract. It would be better to remove the „ethanolic” adjective.

Although the authors in their answer state that this adjective was removed from the text, it’s still present in some instances. If not removing, than maybe change it to „aqueous-ethanolic”? If 90% of the solvent is wate, than the extract is more aqueous than ethanolic.

Response 4: The extraction was carried out with absolute ethanol, using 100 g of plant for every 1000 mL of absolute ethanol. Therefore, it is not an aqueous-ethanolic extract but rather an ethanolic one. However, according to comments requested by the reviewer, the term ethanol was removed from the document. Additionally, modifications were made to the wording and the figure in the method section to clarify the extraction procedure.

Point 5:  Although the part of discussion concerning antioxidant activity was rewritten according to my remarks („there is no trapping of DPPH radical in the assay, it’s scavenged. The same goes for ABTS radical assay”), these expressions should be corrected in subchapter 2.4 and the Experimental part as well. It was not written explicitly in my review of the first version of the manuscript, but it is rather obvious that if it is to be corrected in Discussion, and it’s repeated in Experimental, than it should be corrected in all instances.

Response 5: In the discussion and the experimental section was modified the information indicated by a reviewer on antioxidant effect. Therefore, the term of radical elimination DPPH and ABTS was corrected. The wording of the antioxidant effect of the extract compared to the fractions was modified in the results. The antioxidant effect table was modified. Because a calculation error was detected in the extract data. The units of FRAP (M Fe (II)/ g) were corrected in the table and the methods.

Point 6:  Please add to the text that the fractions were collected by volume (was it 1 mL? Or less?). (subchapter 4.1)

Response 6: In the text, it was added that the fractions were collected by volume of 200 mL.

Point 7: Please add to the Introduction the reason for chosing which bacterial strains were used in this work – as it was explained in the answer to the reviewer’s comments.

Response 7: The reason for choosing the strains used in the study was explained in the introduction.

Point 8: In the experimental part these changes were still not introduced:

- In the formula 1 there is still CN, and it should be NC (negative control).

Response 8: Changes in the negative control (NC) of the formula were made.

Point 9:  Subchapter 4.4: it’s 2,2-Diphenyl-1-picrylhydrazyl, not 2,2-diphenyl-1-picrylhydracil and it’s scavenging, not capture (the latter one was corrected in subchapter 4.4.1, but not in other instances).

Response 9: The correct name was changed from 2,2-Diphenyl-1-picrylhydrazyl.

Point 10: 1A. In the text the name „sinensitin” is used, while in Fig. 2 it’s „sinensetin”.

Response 10: The term sinensetin was changed to sinensitin in figure 2.

Point 11: 2A. As extraction and fractionation are now described together in subchapter 4.1, please change the numbering of following subchapters accordingly.

Response 11: The sequence of the subtitles throughout the document was corrected.

Point 12: 3A. The caption of Table 3 moved to under the table (editing issue).

Response 12: The table edit was corrected to make the title visible.

Point 13: 4A. In Subchapter 2.3.2 there are mentions of Figure 2. However, the results described here are shown in Figure 1.

Response 13: According to the text, the figure number was corrected

Point 14: 5A. Could you give at least some hypothesis why the antioxidant activity of the extract is three orders of magnitude higher than the activity of all fractions? And it is much higher than the activity of e.g. clove essential oil which is known for its strong antioxidant properties (FRAP of 622 micromoles of Fe per gram, compared with 3000 milimoles per gram reported here for the extract). It’s very unusual, and seems more like some calculation error.

Response 14: Corrections were made in the antioxidant effect data of the extract since a calculation error was detected. The concentration of the extract and the fractions employed for calculation were described in the method.

In the discussion, the reason why the extract could have more effect than the fractions was described.

The requested hypothesis is described below:

The extract showed more antioxidant effect than the analyzed fractions. In this sense, F11 had better antioxidant properties than other fractions.

The extraction represents one of the most important factors for the recovery of antioxidant compounds since it is influenced by the solvent, the time and the extraction temperature. However, the chemical composition and physical characteristics of each sample tested are also important factors (Villa-Silva et al., 2020). Given that, after the elution of the extract through the column, some antioxidant compounds could be retained by covalent bonds because some compounds lack affinity for the elution gradient used. Accordingly, the column could be eluted with more polar solvents after elution with 0: 100 Hexane-Ethyl Acetate, Methanol, water, and dilute weak acid solutions are an example because they allow more polar compounds (Mroczek et al., 2020; Antasionasti et al 2017).

Villa-Silva, P. Y.; Iliná, A.; Ascacio-Valdés, J. A.; Esparza-González, S. C.; Cobos-Puc, L. E.; Rodríguez-Herrera, R.; Silva-Belmares, S. Y. Phenolic Compounds of Tagetes Lucida Cav. With Antibacterial Effect Due to Membrane Damage. Bol. Latinoam. y del Caribe Plantas Med. y Aromat. 2020, 19 (6), 580–590. https://doi.org/10.37360/blacpma.20.19.6.41.

Antasionasti Irma, Riyanto S, R. A. Antioxidant Activities and Phenolics Contents of Avocado (Persea Americana Mill.) Peel in Vitro. Res. J. Med. Plants 2017, 11 (2), 55–61. https://doi.org/10.3923/rjmp.2017.55.61.

Mroczek, T.; Dymek, A.; Widelski, J.; Wojtanowski, K. K. The Bioassay-Guided Fractionation and Identification of Potent Acetylcholinesterase Inhibitors from Narcissus C.V. ‘Hawera’ Using Optimized Vacuum Liquid Chromatography, High Resolution Mass Spectrometry and Bioautography. Metabolites 2020, 10 (10), 1–16. https://doi.org/10.3390/metabo10100395.

Point 15: 6A. In Discussion (paragraph  concerning Sytox method) there are contradicting statements: 1) However, some fractions have an effect on S. aureus in ways other than membrane permeabilization. For instance, the sample treated with FM1 did not emit fluorescence but presented MIC90 lower than F3., 2) Therefore, it was confirmed that the antibacterial effect could be due to the increase in permeabilization. So, this paragraph needs rewritting. And it should be stressed out that this results are valid only for S. aureus, as no experiments for other strains has been performed.

Response 15: The paragraph was rewritten as follows:

Nevertheless, only FM1 affects S. aureus other than membrane permeabilization. Because the strain treated with this fraction did not emit fluorescence but presented MIC90 lower than F3, so it has a more antibacterial effect. On the other hand, the membrane permeabilization effect of F3 is similar to that reported by peptides, proteins and phenolic compounds from other plants [45,46]. Therefore, it was confirmed that the antibacterial effect on S. aureus could be due to increased permeabilization [47]. Furthermore, this work showed that bacterial cultures of S. aureus treated with SYTOX Green represent an alternative to measure bacterial viability and susceptibility to antibiotics.

Point 16: 7A. Line 365: why „Surprisingly”? it is quite a common occurence.

Response 16: The wording regarding this expression was modified.

Point 17: 8A. The caption of Figure 3 contains the expression „Graphical abstract”. It’s not a graphical abstract, it’s a scheme.

Response 17: The title of the figure was modified

Point 18: 9A. In FRAP assay there is no radical present, so it can’t be said that The radical scavenging capacity was calculated (lines 481-482).

Response 18: This information was rewritten as described below:

Additionally, ethanol was used as a blank. The FRAP values expressed in micromoles of iron equivalent per dry weight of the material (mM Fe (II) / g).

Point 19: 10A. In the pdf version of revised manuscripts some „μ” in micromoles notation are not visible.

Response 19: Units were reviewed and corrected throughout the document.

Point 20: 11A. Please check carefully if the units are reported everywhere when needed (e.g. when the MIC90s are discussed).

Response 20: Units were reviewed and corrected throughout the document.

Point 21: 12 A. In attached pdf file some instances where grammatical corrections are needed are marked by underlining.

Response 21: Grammar corrections were made, and English language was revised again in the document. For this review, the file marked in blue color indicates the changes made to prevent the Table from moving. Therefore, we prefer not to use the "Track Changes " of Microsoft.

Best regards

The Autors